



# The end of the African humid period as seen by a transient comprehensive Earth system model simulation of the last 8000 years

Anne Dallmeyer[1], Martin Claussen[1,2] , Stephan J. Lorenz[1] and Timothy Shanahan[3]

[1]Max Planck Institute for Meteorology, Bundesstrasse 53, 20146 Hamburg, Germany

[2]Meteorological Institute, Centrum für Erdsystemforschung und Nachhaltigkeit (CEN), Universität Hamburg, Bundesstrasse 55, 20146 Hamburg, Germany

[3]The University of Texas at Austin, Department of Geological Sciences, 1 University Station C9000, Austin, Texas 78712, USA

*Correspondence to*: Anne Dallmeyer (anne.dallmeyer@mpimet.mpg.de)

**Abstract.**

Enhanced summer insolation during the early and mid-Holocene drove increased precipitation and widespread expansion of vegetation across the Sahara during the African Humid Period (AHP). While changes in atmospheric dynamics during this time have been a major focus of palaeoclimate modelling efforts, the transient nature of the shift back to the modern desert state at the end of this period is less well understood. Reconstructions reveal a spatially and temporally complex end of the

AHP, with an earlier end in the north than in the south and in the east than in the west. Some records suggest a rather abrupt end, whereas others indicate a gradual decline in moisture availability. Here we investigate the end of the AHP based on a transient simulation of the last 7850 years with the comprehensive Earth System Model MPI-ESM1.2. The model largely reproduces the time-transgressive end of the AHP evident in proxy data, and indicates that it is due to the regionally varying dynamical controls on precipitation. The impact of the main rain-bringing systems, i.e. the summer monsoon and

extratropical troughs, varies spatially, leading to heterogeneous seasonal rainfall cycles that impose regionally different responses to the Holocene insolation decrease. An increase in extratropical troughs that interact with the tropical mean flow and transport moisture to the Western Sahara during mid-Holocene delays the end of the AHP in that region. Along the coast, this interaction maintains humid conditions for a longer time than further inland. Drying in this area occurs when this interaction becomes too weak to sustain precipitation. In the lower latitudes of West Africa, where the rainfall is only

influenced by the summer monsoon dynamics, the end of the AHP coincides with the retreat of the monsoonal rainbelt. The model results clearly demonstrate that non-monsoonal dynamics can also play an important role in forming the precipitation signal and should therefore not be neglected in analyses of North African rainfall trends.

## 1 Introduction

Periodic variations in the Earth's orbit around the sun trigger the alternation of dry and humid phases in the North African

Sahara (Kutzbach, 1981, deMenocal and Tierney, 2012). The main driver of these changes is the precession of the equinoxes,



which leads to a shift in the time of Perihelion and thus alters the seasonal insolation. Wet conditions return roughly every 20000 years in the Sahara (Skonieczny et al., 2019), as a consequence of increased summertime insolation. The last wet phase established during early- and mid-Holocene and is commonly called the Green Sahara or African Humid Period (AHP, Claussen et al., 2017). The climate and environmental changes during that time are well-documented by diverse proxies from terrestrial and marine palaeorecords. Pollen-based vegetation reconstructions indicate a widespread northward migration of the steppe and savanna biomes with tropical plants occurring as far as 23°N during the AHP (e.g. Jolly et al, 1998; Hély et al. 2014). The Sahara was nearly completely covered by vegetation, supported by substantially increased rainfall (e.g. Hoelzmann et al., 2000; Peyron et al. 2006; Bartlein et al., 2011; Tierney et al., 2017). Permanent lakes spread over the entire North African continent and reached at least up to 28°N (COHMAP Members, 1988; Street-Perrott et al., 1989; Hoelzmann et al., 1998, Tierney et al., 2011; Lézine et al., 2011). Sediment cores taken off the North West African coast characterize the AHP as an interval of low atmospheric mineral dust fluxes from the Sahara, reflecting the overall increase in vegetation and lake area in this region (deMenocal et al. 2000, Adkins et al. 2006, McGee et al., 2013). More humid conditions and the expansion of vegetation also favoured the migration of human populations and cultural development across North Africa (Hoelzmann et al. 2002, Kuper and Kroepelin, 2006).

These changes in environmental conditions are closely related to an enhanced West African summer monsoon system, which was driven by an increased land-ocean thermal gradient resulting from increased boreal summer insolation (Kutzbach, 1981). Furthermore, a number of modelling studies have suggested that precession-driven changes in summer insolation not only intensified the monsoon but also directly influenced the northward extent of the monsoon rainbelt (e.g. Tuenter et al., 2003; Braconnot et al., 2008; Bosman et al. 2014). However, the effects of summer insolation changes were not limited to changes in the monsoon westerly winds (e.g. Su and Neeling, 2005). For example, the African Easterly Jet (AEJ), a strong wind band with core around 600 hPa and 15°N during present-day summer forms due to the strong vertical easterly shear induced by the temperature and soil moisture gradients between the Sahara and lower latitudes (Cook, 1999). The weakening of these gradients during early- and mid-Holocene, lead to a deceleration and northward shift of the AEJ, enhancing the moisture content over North Africa (Patricola and Cook, 2007; Bosman et al., 2012, Rachmayani et al, 2015). The AEJ plays an important role for summer rainfall generation in the Sahel and Sahara. Due to the barotropical-baroclinic instability of the jet (Wu et al., 2012), synoptic disturbances are initiated and organized along the jet, the so called African Easterly Waves (AEWs). Within the AEWs, squall lines and mesoscale convective systems form, which are responsible for most of the annual precipitation in large parts of Northern Africa (Nicholson and Grist, 2003; Skinner and Diffenbaugh, 2013 and references therein; Janiga and Thorncroft, 2016). The dust reduction and the decrease in AEJ speed resulted in less AEW activity during early and mid-Holocene, albeit the latent heat and convection was enhanced (Gaetani et al., 2017). In the upper troposphere, the increase in summer insolation modified the large-sale atmospheric temperature gradients and led to a northward shift of the subpolar/subtropical westerly jet and an intensification of the Tropical Easterly Jet (TEJ) at early- and mid-Holocene (Gaetani et al., 2017). How changes in the TEJ affect the North African rainfall is currently under discussion.



On interannual and decadal timescales, Sahel rainfall and TEJ intensity are positively correlated (Grist and Nicholson, 2001, Nicholson, 2009). This correlation is not present on intraseasonal and synoptic timescales (Lemburg et al. 2019).

A number of atmospheric modelling studies have shown that the pure insolation forcing is not sufficient to reproduce the reconstructed rainfall increase in North Africa (e.g. Joussaume et al, 1999; Braconnot et al., 2000). Feedbacks between the ocean, the land surface and the atmosphere substantially amplify the rainfall response (e.g. Claussen & Gayler, 1997; Kutzbach and Liu, 1997; Zhao et al., 2005; Braconnot et al. 2007, Vamborg et al., 2011; Swann et al, 2014; Rachmayani et al., 2015). Nevertheless, fully coupled Earth System Models still underestimate the intensification and northward expansion of the rainbelt compared with proxy data during the AHP (e.g. Braconnot et al, 2012; Harrison et al, 2014; Perez-Sanz et al., 2014). The suggested causes for this mismatch are diverse. Beside shortcomings in the parametrization and representation of the mesoscale (subgrid-scale) convection and biases in the ocean temperature (Roehrig et al., 2013), the poor representation of land surface properties (e.g. Levis et al. 2004; Vamborg et al. 2011), lake and wetland expansion and their feedbacks on the monsoon system have been proposed to explain model-data discrepancies (Krinner et al. 2013). A possible influence of changes in the dust flux is controversially discussed (e.g. Pausata et al. 2016). However, to date, all of these shortcomings have been  discussed primarily in the context of changes in summer monsoon dynamics. Changes in the atmospheric circulation outside the monsoon season are often ignored. Analyses of the current precipitation distribution show that a large proportion of precipitation in the Sahara is associated with extratropical troughs during spring and fall that penetrate in the lower latitudes and transport moisture into the Sahara in form of concentrated water vapor plumes, the so-called tropical plumes (e.g. Knippertz, 2003; Fröhlich et al. 2013). Although these phenomena occur only sporadically in present day climate, they induce locally heavy rainfall, particularly in times (e.g. during late summer) when the monsoon rainbelt (the moisture supplying system) is still close to its most northerly position, while the subtropical jet is moved back to the south (Knippertz, 2003). It is possible, given the changes in large scale atmospheric circulation during the mid-Holocene, that tropical plumes may have played a more significant role in the increase in Saharan precipitation at that time (Geb, 2000; Kutzbach et al., 2014). For example, Skinner and Poulsen (2013) found an increase in (fall season) tropical plume activity during the early- and mid-Holocene and argued that atmospheric conditions were particularly favourable for the extratropical-tropical interactions (ETI) at mid-Holocene because solar insolation peaked in the early fall. This interaction can act as an amplifier for the insolation-induced and monsoon-related variations in Saharan precipitation.

Over the Holocene, summer time insolation decreased and the monsoon retreated southward resulting in the end of the mid-Holocene humid phase over large parts of the continent. A great deal of research has been focused on understanding the differences in the relative abruptness of the drying at the end of the AHP (e.g. Claussen et al. 1999, Rensen et al. 2003; Liu et al. 2006 and 2007, Collins et al., 2017), which appears to have varied substantially across North Africa (e.g., deMenocal et al. 2000; Kroepelin et al., 2008).  However, it has recently been suggested that the onset of drying also varied spatially, with an earlier end of the AHP in the Sahara than in the lower latitudes and an earlier end in the eastern part than in the western part of North Africa (Shanahan et al., 2015). These differences are seemingly incongruous with a simple southward contraction of the monsoon rainbelt, and may reflect the influence of other regionally important precipitation-generating





processes. To address this question  here, we investigate  the end of the AHP in a transient global simulation of the last 7850 years (Bader et al., 2019) that was performed using the comprehensive Earth System model MPI-ESM1.2. This simulation

includes for the first time not only the orbital, atmospheric greenhouse gas or land use forcing, but also forcings on shorter time-scales (volcanic eruptions, spectral solar irradiance changes). The main focus of this study is the analysis of the main atmospheric circulation changes driving the asynchronous termination of the AHP. The assessment of the abruptness of the transition to the dry state is covered by a follow-up study.

## 2 Methods

### 2.1 The transient simulation

In this study a transient simulation (Bader et al., 2019) of the last 7850 years was performed with the comprehensive Earth System model MPI-ESM1.2 (Mauritsen et al.,2019)  The model consists of the atmospheric general circulation model ECHAM6.3 (Stevens et al., 2013) coupled to the land-surface scheme JSBACH3 (Reick et al. 2013) and the general circulation model of the ocean MPIOM (Jungclaus et al., 2013). JSBACH3 includes the soil carbon model YASSO (Goll et

al., 2015), dynamic vegetation, a 5-layer hydrology scheme (Hagemann and Stacke, 2015) and a dynamic background albedo scheme, which has previously been shown to improve the representation of Holocene precipitation change in North Africa (Vamborg et al. 2011). MPIOM furthermore includes the global ocean biogeochemistry model HAMOCC (Ilyina et al., 2013). Through the inclusion of these subsystem models, the full carbon cycle is enabled in MPI-ESM. However, the atmospheric $CO_2$ concentration is prescribed in the experiments regarded here so that the carbon cycle is not fully

interactive.

The atmosphere and land model was configured with a spectral resolution of T63 (approx. 200km on a Gaussian grid) with 47 levels in the vertical. The ocean model was employed in the horizontal resolution GR15 (i.e. 256x220 on a bipolar grid, 12 to 180km) with 64 vertical levels.

The transient simulation starts ca. 8000 years before present (in our simulation here it is 6000 BCE) and was run until pre-

industrial time (i.e. 1850 CE). For easier nomenclature, we define the early mid-Holocene time slice (further referred to 8k) by the climatological mean of the first 100 years of this simulations (i.e. year 6000-5901 BCE) and the PI reference period by the climatological mean of the last 100 years of this simulations (i.e. year 1751-1850 CE). Accordingly, we define the mid-Holocene time-slice (6k) by the climatological mean of the years 4000-3901 BCE etc.

The transient simulation was performed by using the following forcings:

a) orbital-induced insolation changes (Berger, 1978). These mainly impact the seasonal cycle of e.g. the atmospheric temperature and precipitation. At the start of the simulation, Northern Hemisphere summer insolation is near its maximum and declines gradually toward the present in association with the precession of the equinoxes. The seasonality reduces on the Northern Hemisphere and enhances on the Southern Hemisphere over the Holocene.



b) Methane, nitrous oxide, and carbon dioxide concentrations inferred from ice core records (F. Joos, personal

communication; see Brovkin et al., 2019 and Köhler, 2019). The difference in $CO_2$ between start and end of the simulations

is relatively small, amounting to approximately 20 ppm.

c) stratospheric sulfate aerosol injections imitating volcanic eruptions, prescribed from the Easy Volcanic Aerosol (EVA)

forcing generator (Toohey et al., 2017). This forcing is based on data of the GISP2 Greenland ice core (Zielinski et al., 1996)

only and is probably overestimated during some periods due to overloads by Icelandic volcanic eruptions (cf. Bader et al.

2019). Since the volcanic forcing has a minor effect on the vegetation in our simulation we assume that our results and

conclusions are not affected by this overestimated forcing.

d) Spectral Solar Irradiance forcing, includes extrapolated 11-years solar cycle based on sun-spot observations-sets of far

infrared, near infrared and visible radiation (Krivova et al., 2011).

e) New land-use data adopted from Hurtt et al. (2011): This forcing begins 850 CE with a linear transition period (1000

years) starting 150 BC to slowly built-up the land-use.

A detailed description on the transient simulation and the forcings is given in Bader et al. (2019) and Brovkin et al. (2019).

For some analyses, daily output was needed. For this purpose, a few time-slice experiments have been re-run for 30 model

years (Table 1). For these time-slices, periods with low volcanic activity have been selected, representing 7k, 5k, 3k and

0.3k. As the Earth's orbital parameters change over time, the length of the seasons varies for the different time-slices. To

avoid artifacts of the fixed-calendar (i.e. the modern calendar) used in the climate models, daily output has been re-assigned

to months defined by fixed angular on the Earth's orbit following Bartlein and Shafer (2018).

## 2.2 Defining the end of the African humid period in the model

In order to assess the end of the African Humid Period in the transient simulation, we analyse changes in the bare soil

fraction (BSF) assuming that the BSF is an appropriate indicator for the moisture availability in this region. For this purpose,

we strongly smoothed the time-series for each grid-cell by performing a local regression with a smoothing span of 70%

using the LOESS routine in R (R core team, 2014). To define the onset of drying, we determined the first inflection point

($T_{end}$) in the smoothed BSF record where the slope of two consecutive years exceeded the slope of a straight line between the

BSF at 8 k and PI (Fig. 1). To ensure that this drying reflected the end of the AHP, we additionally imposed the constraints

that (i) $T_{end}$ occurs after the minimum in BSF (peak wet conditions) and (ii) that the slope of the BSF curve remains larger

than the straight line between 8 k and PI. Although this approach is relatively simplistic, given the relatively coarse temporal

resolution of the proxy reconstructions, this approach is likely sufficient for comparison with the large-scale patterns in the

timing of drying evident in the proxy data. For grid-cells showing an increasing BSF over the Holocene, the AHP end is

assigned to PI (i.e. the end of the simulation).




### 2.3 Proxy data synthesis

For comparison with the modeling results, we revised and expanded the proxy data synthesis of hydro-climatic changes at the end of the AHP previously published in Shanahan et al. (2015). Our new synthesis includes 138 records from sites across North Africa (see supplement). We revised the data compilation to include additional records published after or not identified in the earlier synthesis. In a few cases, records that were deemed too short to properly identify the decline in precipitation (i.e., because of deflation), were excluded. For each record in the synthesis, we identified the first major decline in precipitation/hydro-climate from the proxy data time series in order to be as consistent as possible with the model output. An exception was made when the initial onset of drying was followed by a substantial return to wetter conditions, indicating that the initial drying was not associated with the end of the AHP. The identification of the AHP end was performed visually because the noisiness of the data made it impossible to use other techniques such as change point analysis on most records.

We excluded marine sites in the model-data evaluation because of the uncertainties associated with comparing the model derived moisture estimates against the marine proxy records, which receive terrestrial contributions (fluvial or aeolian) from potentially distant source regions. We are aware of the fact that the undefined and varying source area also complicates the comparison of the model results and the terrestrial records. While the model mainly shows the local conditions, the reconstructions may be influenced from regional or extra-regional changes (e.g. extra-regional pollen load). However, we assume that the extra-regional influence is less pronounced in the terrestrial records.

Another caveat in the records is the partly very poor dating and the coarse temporal resolution probably leading to additional uncertainty in the assessment of the AHP end time. To better compare the reconstructions with the model results, we therefore round up the AHP ends in the records and in the model to 500year-intervals.

An alternative approach that partly overcomes the above mentioned caveat is to compare the simulated and reconstructed patterns of drying, only. To do so, we estimated the relative timing of the end of the AHP for each grid cell containing a proxy site. By simply summing up the different sites/grid-cells, we determined whether the AHP end was earlier, later or the same as in the other sites and grid-cells.

### 3 Evaluation of the simulation

### 3.1 Precipitation patterns

One of the main issues in current model experiments simulating the mid-Holocene is the strong underestimation of precipitation and of the northward extend of the rainbelt over North Africa compared with reconstructions (e.g. Braconnot et al., 2012). Our simulations improve somewhat upon this discrepancy, but still show some important differences. The simulated mid-Holocene annual mean deviation from the present day precipitation within MPI-ESM is substantially larger than the one inferred previously from modelling studies (i.e., PIMP3) and more consistent with quantitative precipitation estimates from pollen data (Bartlein et al., 2011) (Fig. 2). At the same time, however, our simulations also show higher regional rainfall maxima and a larger range of spatial variability in precipitation than seen in the proxy data. Furthermore, the





median value for mid-Holocene precipitation changes in our simulations is significantly lower than suggested by the proxy

reconstructions (Fig. 2b). Although some of these discrepancies may be associated with the limited spatial coverage in the proxy data, there are some important features in the proxy data that the model fails to capture. Most significant amongst these is the fact that the northward extent of the monsoon rainbelt is still underestimated in our simulations (Fig. 2a). Our simulations also show particularly large precipitation changes in the Sahel (>1000 mm/yr) that exceed existing pollen-based estimates (Bartlein et al., 2011), but are consistent with recent estimates from the hydrogen isotope composition of leaf wax

biomarkers in marine sediments from the same latitudes (cf. Tierney et al., 2017). Regardless, it appears that in our simulations the monsoon rainbelt is trapped between 10° and 20°N, unable to expand northward into the Sahara, in disagreement with the proxy evidence for wetter conditions throughout the Sahara during the mid-Holocene. Only the northwestern-most Sahara (west of 0°) received much more rainfall during the mid-Holocene, with changes of up to 200mm/year in that region. As a consequence of the limited northward shift of the rainbelt in the MPI simulations,

precipitation is presumably underestimated over most of the Sahara and overestimated in the Sahel compared with reconstructions.

## 3.2 Vegetation distribution

Fig.3a and 3b show the reconstructed and simulated PI and 6k vegetation distributions in form of mega-biomes that have

been calculated on the basis of the simulated plant functional type fractions and respective bioclimate, following the method of Dallmeyer et al. (2019). The simulated PI extent of the various mega-biomes agrees well with modern reconstructions (BIOME6000, Harrison, 2017) except for equatorial East Africa where the model overestimates the desert area. As expected, vegetation cover is strongly increased over most of north Africa at mid-Holocene. The steppe biome is shifted northward by up to 10°, leading to a large expansion of the savanna and tropical forest belts. Grasslands are more widespread along the

northwestern Saharan coast, reaching far onto the continent. In the western Sahara, the simulated change in the grassland biome pattern strongly matches the BIOME6000 reconstructions but the extent of the steppe is underestimated with respect to the northern and southern boundary in the central Sahara. According to reconstructions, large parts of the Sahara were vegetated (Jolly et al. 1998), but the model indicates that the northward shift in the desert-steppe border only reached 22°N. Interestingly, almost every grid cell in the model was at least to 10% covered by vegetation at some point during the African

Humid Period (Fig.3c). Only a small region in North-East Africa was desert during the entire 8000 years of simulation. This Holocene minimum in desert extent agrees well with the extent of the vegetation derived for 6k in reconstructions, indicating that at least for some time during the Holocene the model shows a similarly strong vegetation change as the reconstructions. In total, the model shows strong vegetation changes during the last 8000 years. The vegetation cover in North Africa has almost halved from 8k to PI. While in the mean vegetation varies smoothly over the course of the Holocene, individual grid-

cells suggest heterogeneous vegetation dynamics with different starting times of the vegetation decline (Fig.3c).



## 4. Results and Discussion

### 4.1 Proxy-model comparison of the African Humid Period

In agreement with the updated proxy data synthesis, the model indicates a time-transgressive end of the AHP (Fig.4). The wet period ends earlier in the North than in the South and earlier in the East than in the West. In addition, the model shows significant differences in the timing of drying between the Eastern and Western Sahara. However, these differences are not captured in the reconstructions because of the low spatial density of records in this region. As expected, due to the low temporal resolution of the proxy data and model biases, the kappa metric (Cohen, 1960) reveals only a poor match between

the simulated AHP end periods and the reconstructions (κ=0.03). However, we note that this method has a tendency to underestimate the agreement in maps which are offset against each other (Foody, 2002; Tang et al., 2009).

There is much better agreement between the model and proxy data comparing the pattern of the relative timing of the AHP termination (64%, κ=0.42, Fig.4). This suggests that while the absolute timing of drying in the proxy and model data do not match well, the spatial patterns in the differences in the onset, with some areas drying earlier and others drying later, is

reproduced by the simulation. With few exceptions, the mismatched sites are all located in Eastern Africa (east of 30°E) and along a 5° of latitudes-tilted band at the modern Sahara-Sahel boundary, where the model indicates that the AHP ended later than suggested by the reconstructions. This region of disagreement coincides with the transition from the area where precipitation is rather overestimated (roughly south of 18°N) to where it is underestimated (Sahara) at 6k. We infer that the discrepancies thus reflect regional biases in the model simulations of the northward extent of the monsoon rainbelt; the

rainbelt remained in this region too long before retreating to its current position in the late Holocene.

In Eastern Africa, the model suggests an earlier end of the humid period for most sites. At least part of this mismatch may be caused by the complex orography in Eastern Africa which is only poorly represented in the coarse resolution of the model. Therefore essential regional circulation systems (such as the Turkana jet) are missing or at least poorly represented in the model. Furthermore, the coarse resolution orography could impede the simulation of the Indian monsoon development. On

the other hand, some terrestrial reconstruction may rather portray the local changes in climate instead of the regional trend.

### 4.2 Rainfall regimes and the time-transgressive termination of the African Humid Period

Vegetation changes in Northern Africa are highly coupled with changes in available moisture. To understand the causes of the time-transgressive end of the AHP, a detailed analysis of the precipitation changes is thus indispensable. Fig. 5 shows the

onset pentad (i.e. intervals of 5-days) and length of the rainy season for the 7k time-slice simulation (cf. Tab.1), calculated on the basis of the definition by Wang and Linho (2002). Similar as for the AHP end, both patterns indicate a north-south and east-west gradient. The rainy season begins earlier and lasts longer in the south than in the north and begins earlier and lasts





longer in the west than in the east, which is to a large part related to the seasonal march of the West African summer monsoon penetrating inland from the Gulf of Guinea. Only on the Ethiopian Plateau, the rainy season also starts very early in
the year. In the region north of 7°N, the end of the AHP is negatively correlated with the onset of the rainy season at 7k (r=-0.47) and strongly positively correlated with the rainy season length (r=0.6). The length of the rainy season thus already explains more than 1/3 of the spatial variability of the AHP end. This underlines that the timing of the AHP end is strongly connected to the rainfall changes or, more precisely, with changes in the number, the extent and the duration of the strong rain events bringing most of the precipitation.

We are aware of the fact that the time-slice 7k is only a snapshot, but in principle the pattern looks very similar in other time-slices, although the region receiving precipitation is of course shrinking throughout the Holocene. Additionally, it is reasonable to assume that the regions experiencing the latest onset and shortest rainy season at 7k are also the regions in which the rainy systems decrease and disappear first in the course of the Holocene.

Although much of North Africa is strongly influenced by the West African summer monsoon, the impact of the monsoon
circulation on seasonal rainfall can differ significantly, depending on location. For example, the monsoon rainy season starts earlier near the southern coast, ends later, and is associated with drier conditions in July and August, when monsoon rainfall is at a maximum over the Sahel. Furthermore, northwestern Africa (> 20 N, < 20 E) and occasionally also northeastern Africa are also affected by extra-tropical troughs developing in the subtropical jet, which can advect significant amounts of tropical moisture into the Sahara in the form of concentrated plumes of water vapour (Knippertz, 2003 and 2007). Fig. 6
shows the typical daily-mean precipitation and upper-tropospheric wind pattern for a strong rain event in the Sahara, based here on a day in October taken from the 7k time-slice simulation. Well displayed is the Rossby-wave train embedded in the strong subtropical jet forming an upper-lever trough with a SW to NE orientated axis that reaches far into the low latitudes. On the Eastern flank of the trough, a broad cloud band develops (so called tropical plume, not shown) extending from the tropics to the Mediterranean. The tropical plume coincides with abundant rainfall that is further favoured by the divergence
in the outflow of the jet streak and the related positive vorticity advection (not shown). Lower tropospheric air can thus be transported rapidly into the subtropical upper troposphere (e.g. Froehlich, 2013 and references therein). The rainfall for this event exceeds 25 mm/day in some parts of the Sahara. In general, rain events associated with these tropical plumes usually provide more than half of the mean annual precipitation at present, sometimes within only a few days, leading to widespread flooding and destruction (Knippertz and Martin, 2005). Tropical plumes are therefore a factor that should not be neglected in
rainfall analysis for North Africa.

The differences in the timing and the sources of precipitation result in distinct regional climatologies. Here, to better distinguish these regional differences in rainfall seasonality, we performed a c-means clustering (Meyer et al., 2014) based on pre-industrial monthly-to annual rainfall and identified the following distinct rainfall regimes (Fig. 7):

a) equatorial zone (yellow): year-round precipitation with a double rainfall peak (spring and autumn) related to the seasonal
advance and the retreat of the monsoon rainbelt over the region.

b) monsoon domain (orange): typical monsoonal seasonal cycle with a single rainfall maximum in summer (August).





c) westernmost Sahara (red): contributions from both the summer monsoon and extratropical troughs, which produce an extended rainfall peak in August and September.

d) northwestern Sahara (green): autumn and winter precipitation, strongly effected by ETIs with peak precipitation in
October/November.

e) Eastern Sahara (blue): limited wintertime precipitation, peaking in December/January.

We hypothesize that the change in the orbital forcing throughout the Holocene affects these rainfall 'regimes' differently, leading to regional differences in the timing of the end of the AHP. The Holocene orbital forcing induces mainly a seasonal signal as the changes in insolation mostly compensate each other in the annual mean. Due to the different seasonal rainfall
cycles, modifications in the dynamics of the key players (i.e. summer monsoon and ETI) should result in different responses to the orbital forcing in the respective regions.

### 4.3 Changes in the dynamics of the major rainfall-bearing atmospheric circulations

### 4.3.1 The West African monsoon system

At 7k, the summer monsoon related inflow from the Atlantic ocean appears as a broad, zonally orientated wind belt reaching
up to 18°N that turns north towards the Red Sea at longitudes east of 20°E (Fig.8a). The moisture flux onto the continent and into the Sahara is much stronger at 7k than during the 0.3k time-slice. Convergence occurs over a significant portion of North Africa (ca. 0-25°N), reflecting the expanded latitudinal range of the monsoon rainbelt during the mid-Holocene. The exception to this is the Eastern Sahara where precipitation remains low because of low-level moisture divergence at this time. The boundary between moisture convergence and divergence is inclined by approx. 20° NW-SE, representing the
inclined rainband. As a result, gradients at the AHP end are already manifested in a 'tilted' monsoon system.

The general pattern of moisture convergence changes only slightly during the Holocene, instead showing a weakening trend that is consistent with the gradual decline in insolation forcing. The reduction in monsoon strength can be described by a simple monsoon index, which we define here as the relative change in the product of moisture and zonal wind speed at 850hPa (=q·u), averaged over the western Sahel (10W-0E, 8-18N). This zonal moisture inflow is more than 6-times stronger
at 8k than at PI (Fig. 8b) and decreases relatively linearly, but more intensively from 8k to 3k than from 3k to PI. The decline in monsoon strength is accompanied by a reduction in the area affected by monsoon rains. The retreat of the main monsoon rainband is reflected by the bipolar convergence change in the monsoon domain, indicating a weakening of the low level moisture flux convergence in the north and a strengthening of the flux in the south of the rainbelt between two time-slices (Fig. 8a). This pattern is most prominent between 5k and 3k.

Fig. 9 displays the change in the extent of the monsoon rainbelt, based on the 2 mm/day precipitation isohyet in the annual mean. These isolines are tilted with a similar angle as the time-isolines of the AHP end. The monsoon retreat occurs meridionally differently. In the most western part of the Sahel (west of 10W), the rainbelt remains at its northern position for several thousand years, the strong retreat doesn't start until 4k. Here it is noticeable, that the monsoon rainbelt retreats more



slowly in the early mid-Holocene than in the late Holocene. The spatial variability in the timing of the AHP end in the main
monsoon domain strongly correlates (r=0.84) with the position of the 2 mm/day isolines.

Summarizing, the monsoon system gradually weakens during the Holocene, leading to a southward retreat of the monsoon
rainbelt, in line with the generally accepted orbital monsoon hypothesis (Kutzbach, 1981). The moisture flux reveals that the
monsoon decline coincides with a strengthening of the low-level moisture flux convergence in the southern part of the
monsoon domain and a weakening of the convergence at the northern boundary. This underlines why the AHP ends first in
the northern part, where the moisture sink is weakened and remains in the southern part, where it is further enhanced. The
AHP end in the main monsoon region can be explained fully by the retreat of the West African summer monsoon.

### 4.3.2 The impact of changes in extratropical troughs

To get an overview of the change in strong rain events associated with extratropical troughs during the Holocene, we analyse
all rain events with daily rainfall exceeding 4 mm/day occurring in the Sahara within the 30 years of the individual time-
slices simulation, following Skinner and Poulsen (2013).

Fig. 10 shows the number of rain events (> 4mm/day) in the Sahara at 7k and the change in this number during the
Holocene. At 7k, the Western Sahara (10°W-10°E, 20-35°N) experiences more rain events compared to 5k, while in the
region along the western Atlantic coast (15-11W, 20-25°N, further referred to as coastal area) rain events show the tendency
to increase from 7k to 5k. This is the region indicating a pronounced delay in the timing of the AHP end. From 5k to 3k and
from 3k to 0.3k rain events reduce strongly in both regions with increasing magnitude from the North-east to the South-west.
What are the atmospheric dynamics resulting in these regionally different trends between the coastal area and the central
Western Sahara? To answer this question, we need to look again at the special dynamics associated with rain events in
Northern Africa. Fig. 11 shows the vertically integrated moisture flux anomalies during 7k September and October for
composites of all rain events occurring in the Western Sahara and the coastal area, respectively, in comparison with the
monthly mean flow. Rain events are not unusual in the coastal region during September. The composite mean differs only
slightly from the monthly mean flow, showing just a little increased moisture flux convergence in Mauritania and at the
Moroccan coast. This anomaly is mainly related to a slightly stronger, more northward turning monsoon flow and a more
inclined AEJ at the coast compared to the monthly mean (for details see Appendix A), resulting in an intensified moisture
convergence there.

During October, these anomalies in the lower levels are much more pronounced. In addition, a slightly cooler/warmer upper
troposphere above the Atlantic/Eastern Sahara (centre ca. 5°W 27°N) triggers south winds along the northern coast.

Fig. 11 underlines that strong rain events in the Western Sahara not only coincide with a strong meridional moisture transport
from low latitudes to the Mediterranean Sea but also with an abnormally strong monsoon-like inflow from the Atlantic,
stretching deep onto the continent, independent of the month in which the events are occurring. The moisture flux diverges
slightly more intensively over the Eastern Sahara and converges distinctly over the Western Sahara compared to the mean.
This region of enhanced moisture convergence fits very well to the region in which the AHP end is delayed (i.e. later than





8k). Interestingly, the occurrence of rain events in the Western Sahara seems to be unfavourable for precipitation along the western coast (and in the coastal area), as the moisture convergence is reduced there.

This pattern is at least partly related to changes in the AEJ (cf. Fig. A2) that is substantially reduced in its outflow domain, diminishing the moisture transport to the coast. Beside the circulation anomalies coinciding with the strong equatorward extending trough in the upper troposphere that portray the classical pattern of an tropical plume situation (Fig.A1), heavy rain events in the Western Sahara are associated with a strong cyclonic, monsoon-like anomaly in the lower troposphere with strong south winds around 0°E (Fig A3).

To summarize, the atmospheric conditions for heavy rain events in the coastal area are part of the normal late-monsoon circulation during September. This 'mean' circulation is 'maintained' during October, when abundant rain falls in this region. The subtropical jet meanders slightly and supports the formation of precipitation along the coast. In contrast, rain events in the Western Sahara are related with a pronounced Rossby-wave anomaly with cold air above the Western coast and the Atlantic, triggering a south wind anomaly in the central western Sahara throughout the entire troposphere. The position of this cold air anomaly determines whether rain falls along the coast (i.e. in the coastal area) or in the Western Sahara. In the latter case, the anomaly is much more pronounced and centered on land, while in case of heavy rain events at the coastal area, the anomaly is centered about 10° further westward, above the Atlantic.

During the course of the Holocene, the upper atmosphere experiences a cooling (decrease in geopotential) in September and October due to the decreasing insolation (Fig 12). As the cooling is generally more pronounced in the northern latitudes than near the equator, the upper level westerlies are continuously enhancing and the subtropical jet penetrates more and more towards the Sahara. Therefore, the upper tropospheric conditions for the formation of rain events in the Sahara are generally improving during the Holocene.

During October, the upper level temperature above the northern Sahara and Mediterranean regions strongly decrease from 7k to 5k. This leads to a cyclonic anomaly centered at the Libyan coast and enhanced north winds above the Western Sahara, resulting in mean atmospheric conditions that rather prevent the moisture from being transported out of the lower latitudes, favouring particularly the rain events at the coast. From 5k to 3k upper tropospheric cooling in October is most pronounced above Spain and Morocco, resulting in a wind anomaly with anomalous south winds along 0°E.

In the mid-troposphere, the warming of the continental interior (probably related to less precipitation and evaporative cooling) leads to a decrease in the meridional temperature gradient towards 5k and a slight weakening and southward displacement of the African Easterly Jet (not shown). Directly at the coast, the wind is blowing from the south slightly more, favouring the coastal rain events rather than Saharan rain events at 5k. In the following millennia, mid-tropospheric temperatures in North Africa increase strongly so that the easterly flow is substantially reduced and the African Easterly Jet moves far to the south.

As discussed in Sect. 3.3.1, the West African monsoon is continuously weakening and retreating to the south during the Holocene. The changes in vertically integrated moisture flux displayed in Fig. 7 reflect the changes in the position of the monsoon rainband. They indicate that monsoonal moisture flux convergence during summer is enhanced in the coastal





region at 5k compared to 7k. West wind reaches still up to 15°N in October and the low level wind field at the coast has a more pronounced southward component at 5k compared to 7k (Fig.13), additionally favouring rain events in the coastal region. From 5k to 3k, the low-level temperature and with it the wind field change much more strongly. The monsoon is located far in the south and is no longer relevant for the North African atmospheric dynamics during October.

To summarize, the changes in the upper and mid-tropospheric dynamics are generally providing conditions that are getting more favourable for the formation of rain events in both, the coastal region and in the Western Sahara, towards the late Holocene. The upper level westerly winds penetrate deeper onto the North African continent, likely enhancing the probability of extratropical troughs that extend into lower latitudes. At 5k during October, the distinct temperature change in the Mediterranean region hampers the moisture transport to the Western Sahara. The African Easterly Jet and the monsoon

flow is continuously decreasing and moving southwards during the Holocene. However, the shifting of the Perihelion towards autumn during mid-Holocene favours the intensification and maintenance of the (late) monsoon circulation. This is in line with the results of Skinner and Poulsen (2013). Due to the 'tilted' structure of the monsoon system, tropical moisture can be transported deeper to the north along the western coast than in the eastern part, so that the low-level moisture flux to the coast decreases only slightly from 7k to 5k. The African Easterly flow and the monsoon are still present in the coastal

area at 5k and thus have the chance to interact with the upper tropospheric circulation. The interplay between a still strong and active monsoon circulation and the intensified upper-level westerlies tend to increase the coastal rain events in October during 5k compared to 7k (Fig. A4), contributing to the delay in the AHP end. In the following millennia, the monsoon dynamics and the extratropical troughs are spatially developing into distinct phenomena, i.e. the extratropical and tropical circulations are 'decoupled'. Their interaction is weaker and therefore less common after 5k. The number of rain events in

both regions is reducing substantially during the late Holocene.

**5 Summary and Conclusion**

For several decades, the African Humid Period has provided a challenging palaeoclimate modelling target due to the difficulties in reproducing the magnitude and extent of wet conditions suggested by proxy data and the evidence for

nonlinear behaviour during the transitions into and out of the AHP, both of which have been ascribed to the impact of land surface feedbacks on the monsoon. More recently, Shanahan et. al (2015) argued that the end of the AHP was time-transgressive rather than uniformly abrupt, and that regional-scale differences in the seasonality of monsoon rains could explain this variable monsoon response to the slowly evolving insolation forcing. Here, using high-resolution transient simulations of the last 7850yr from the comprehensive Earth System Model MPI-ESM1.2, we calculated the spatial

evolution of the AHP termination across North Africa. Despite some differences in the spatial extent and magnitude of precipitation changes during the AHP, particularly over the Sahara, our experiment is in broad agreement with the spatial patterns in the timing of the AHP end in the proxy data. In general, the end of the AHP began earlier in the north than in the south and occurred earlier in the east than in the west.





As North Africa is not only affected by the (tropical) monsoon circulation but also by extratropical troughs penetrating into
lower latitudes and transporting moisture towards the Sahara, the continent regionally faces very different seasonal cycles in
rainfall. We hypothesized that the (seasonal) changes in the insolation forcing throughout the Holocene affect these 'key-
players' and the regional rainfall 'regimes' differently, causing the spatial variations in the timing of the end of the AHP.
Based on the seasonal rainfall cycle, five regions were distinguished here (see Fig. 7a)

a) Equatorial zone (yellow), region with year round precipitation:

In the Eastern part of this zone, reconstructions and model results deviate. The AHP end is rather 'patchy' which may be
related to the fact that the signal is very weak and not robust. Furthermore, the orography in this zone in very complex and
under-represented in the model. A distinct reason for the pattern in AHP end can not be derived from the model. In the
western (coastal) part, rainfall increases during the Holocene due to the southward shift of the monsoon rainbelt. The AHP
ends late or is still present in pre-industrial times.

b) Main monsoon domain (orange), region that is solely affected by the summer monsoon:

The AHP end in the main monsoon domain can fully be explained by the retreat of the West African summer monsoon
(r=0.84). The monsoon circulation is gradually weakening and shifting southwards during the Holocene. As the monsoon
rainband is meridionally tilted, the rainband stays longer in the western part than in the eastern part, imprinting the east-west
gradient on the AHP end.

c) Westernmost Sahara (red), region with summer monsoon rainfall and rainfall related to extratropical troughs:

During early- and mid-Holocene, the monsoon season is prolonged to September, leading to strong moisture supply in the
Western coastal area. Due to the shift in Perihelion towards autumn, the maintenance of the monsoonal circulation pattern
into October is favoured at mid-Holocene. The upper-level westerlies intensify during the Holocene, providing condition that
foster extratropical-tropical interactions. Due to the interplay of a still strong monsoon and a strengthened subtropical jet,
rain events at the coast increase from 7k to 5k, leading to a delay in the AHP end. Afterwards the circulation systems become
decoupled and the extratropical-tropical interaction reduces.

d) Northwestern Sahara (green), region with autumn and wintertime precipitation, strongly effected by ETIs:

The number of rain events decreases gradually during the Holocene (here since 7k) as the moisture supply is reduced due to
the equatorward retreat of the monsoon system. The interaction of the extratropical and the tropical circulation gets weaker,
since the circulation systems get more and more decoupled. As the pre-conditions for the formation of rain events are getting
more favourable closer to the western coast, rain events can 'survive' longer in that region. The rainfall surplus due to ETIs
vanished first in the central northern Sahara and than in the western part, leading to the a slight gradient in the AHP end in
the Western Sahara.

e) Eastern Sahara (blue), region with (little) wintertime precipitation:

This region neither profits from monsoonal rainfall nor from the ETIs, precipitation is rare and does not change much during
the Holocene. According to the model., no humid phase occurs during the entire Holocene in this region





We explain the reconstructed time-transgressive end of the African Humid Period based on a transient experiment performed in a comprehensive Earth System Model. The model results clearly show that the regionally different seasonality affects the response of the rainfall to the Holocene insolation change. The mid-Holocene atmospheric conditions are optimal for the initiation of rainfall events in the western Sahara via extratropical-tropical interactions, as the monsoon season is prolonged and the subtropical westerly jet can reach far to the south during the late summer and fall season. In the course of the Holocene, the monsoon weakens and moves southward, while the upper level subtropical westerly jet intensifies. Both changes compensate each other for several thousand years until the extratropical and tropical circulation become decoupled. Due to this interaction, the AHP end is delayed in the western Sahara. Humid conditions are maintained longest at the southwestern Saharan coast, where the combined effect of enhanced monsoon and increased extratropical trough occurrence favours precipitation most (at least up to 5k). In the main monsoon region the asynchronous end of the AHP is solely controlled by the retreating monsoon system. As the monsoon rainband is tilted zonally, the AHP ends earlier in the east than in the west (on the same latitude). Regions in the Sahara to which the moisture is transported neither by extratropical troughs nor by the monsoon flow, show an AHP end directly at the beginning of the simulations (i.e. there is no AHP).

Our results show that the analysis of rainfall trends in North Africa should not be limited to the monsoon season. Insolation changes affect the atmospheric circulation year-round and the impact of the non-monsoonal processes may vary with time, also with respect to their contribution to the annual rainfall signal. Thus, non-monsoonal processes may be important in other climate states, even if they are less relevant in today's climate. Our results furthermore raise the question whether the abrupt transition into the drier state recorded in sediment cores off the western Saharan coast must also be interpreted in the context of a decoupling of the extratropical and tropical atmospheric circulation somewhere during the mid- to late Holocene that may have led to a regionally relatively fast termination of the humid period.

## 6 Code and data availability

The relevant data, all scripts used for the analysis and supplementary information that may be useful in reproducing the authors' work will be archived by the Max Planck Institute for Meteorology and can be obtained by contacting publications@mpimet.mpg.de





**Appendix A: Circulation anomalies associated with strong rain events in the Sahara**

Fig A1-A3 show the 7k September and October mean atmospheric circulation in 300hPa (tropical and subtropical jet level), 600hPa (AEJ level) and 850hPa (monsoon level) and the anomalies in this circulation coinciding with strong rain events (composite mean).

Heavy rainfall events in the coastal area characterize the September mean moisture flux during 7k. The composite mean of all rain events (>4mm/day) differs only slightly from the monthly mean flow, showing just a little increased moisture flux convergence in Mauritania and at the Moroccan coast (cf. Fig. 11). The moisture flux is determined by a strong easterly wind band between 20-30°N, which can partly be explained by a Tropical Easterly Jet extending further north (Fig A1). On the other hand, a cold air anomaly directly at the coast (centre at 20W and 20N), which reaches far into the troposphere, leads to a slight northward inclination of the African Easterly Jet in its outflow (Fig A2). This favours vertical uplift at the coast and gives the atmospheric flow a small southerly wind component at the jet level. The cold air anomaly associated with rain events at the coast furthermore causes a slightly stronger and more northerly extending monsoon flow, which has a pronounced, albeit regionally confined, south wind component (Fig. A3).

In contrast, October rain events at the coast coincide with a pronounced monsoon-like moisture flux onto the continent in the region 8°-18°N (west of 5°W) turning to the north and resulting in an intensified moisture convergence at the coast (north of 18°N) compared to the mean state. This southerly wind anomaly runs through the entire troposphere. In upper levels (here: 300hPa, Fig 12), a Rossby wave like anomaly with higher geopotential above the northwestern Sahara and lower geopotential above the Atlantic leads to an anticyclonic flow (centre ca. 5°W 27°N) triggering southwind along the coast. This temperature anomaly furthermore results in a tilting of the African Easterly Jet axis towards the Canary Islands so that anomalous southerly winds can establish also at this level. The low-level atmospheric flow is characterised by a monsoon-like inflow and a northward flow along the coast. To summarize, the atmospheric conditions for heavy rain events in the coastal area are part of the normal late-monsoon circulation during September. During October, this circulation is 'maintained' when abundant rain falls in this region.

Strong rain events in the Western Sahara are associated with a strong meridional moisture transport from low latitudes to the Mediterranean Sea and an abnormally strong monsoon-like inflow from the Atlantic stretching deep onto the continent. The moisture flux converges distinctly over the Western Sahara and diverges slightly more intensively over the Eastern Sahara and along the western coast compared to the mean (cf. Fig.11). This pattern is at least partly related to the changes in the African Easterly Jet (A2) that is substantially reduced in its outflow domain, diminishing the moisture transport to the coast. Lower geopotential height along the north western coast leads to a cyclonic circulation anomaly with strong south winds around 0°E. This anomaly extends through the entire troposphere and results in a monsoon-like inflow (FigA3) and southwest winds in 850hPa which reaches further inland than in case of coastal rain events. In the upper levels (Fig.A1), a large trough forms, penetrating deep into the Subtropics, portraying the classical pattern of the extratropical – tropical interaction. The atmospheric circulation anomaly for Sahara rain events is similar in September and October, although it is



more pronounced during October. The region of moisture convergence fits very well to the region in which the AHP end is delayed (i.e. later than 8k). On the other hand, the regions in the Sahara to which the moisture is transported neither by extratropical troughs nor by the monsoon flow, show an AHP end directly at the beginning of the simulations (i.e. there is no AHP).

**Author contributions**

AD and TS wrote the manuscript, SL performed the simulations, MC and AD planed the study. All authors discussed the analysis and the manuscript.

**Competing interests.**

The authors declare that they have no conflict of interests.

**Acknowledgements**

This work contributes to the project PalMod, funded by the German Federal Ministry of Education and Research (BMBF),
Research for Sustainability initiative (FONA, www.fona.de). AD was financed by PalMod. We thank T. Kleinen (MPI-M) for his helpful comments on an earlier version of this manuscript.



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





| Time-slice acronym | Model years | Start year, Relative calendar | Period, Gregorian calendar |
|---|---|---|---|
| 8k | 1001-1100 | 7999 BP | 6000-5901 BC |
| 7k* | 1961-1990 | 7039 BP | 5040-5011 BC |
| 6k | 3001-3100 | 5999 BP | 4000-3901 BC |
| 5k* | 4001-4100 | 5000 BP | 3001-2971 BC |
| 3k* | 6001-6030 | 2999 BP | 1000-971 BC |
| 0.3k* | 8701-8730 | 299 BP | 1700-1729 CE |
| PI | 8751-8850 | 249 BP | 1750-1849 CE |


Table 1: List of time-slices used in this study. For the time-slices marked with *, simulation has been restarted to produce daily output for 30 model years. Please notice that an interval of low variability had been chosen for these simulations, therefore, the time-slice 7k actually starts before 7k (at model year 1961 instead of model year 2001). Please also notice that all means of theses time-slices are based on 30 years only, while the means on 8k, 6k and PI are based on 100 years. BP is

here defined by years before the year 2000CE.





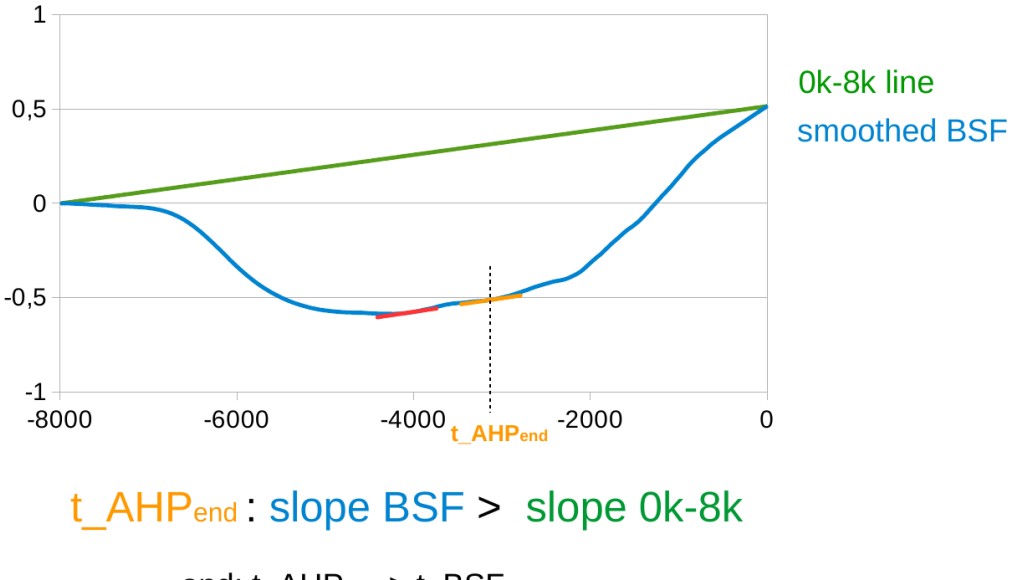

Fig. 1: Sketch of the definition of the AHP end in the model. We calculate the AHP end based on the strongly smoothed (loess filter, span = 70) bare soil fraction (BSF) change. The time step $T_{end}$ at which the slope between two consecutive time steps exceeds the slope of a straight line between the 0k and the 8k bare soil fraction is regarded as end of the AHP. Additionally, the minimum of the bare soil fraction has to precede $T_{end.}$ and after 500 years, the slope has still to be larger than the slope of the straight line.





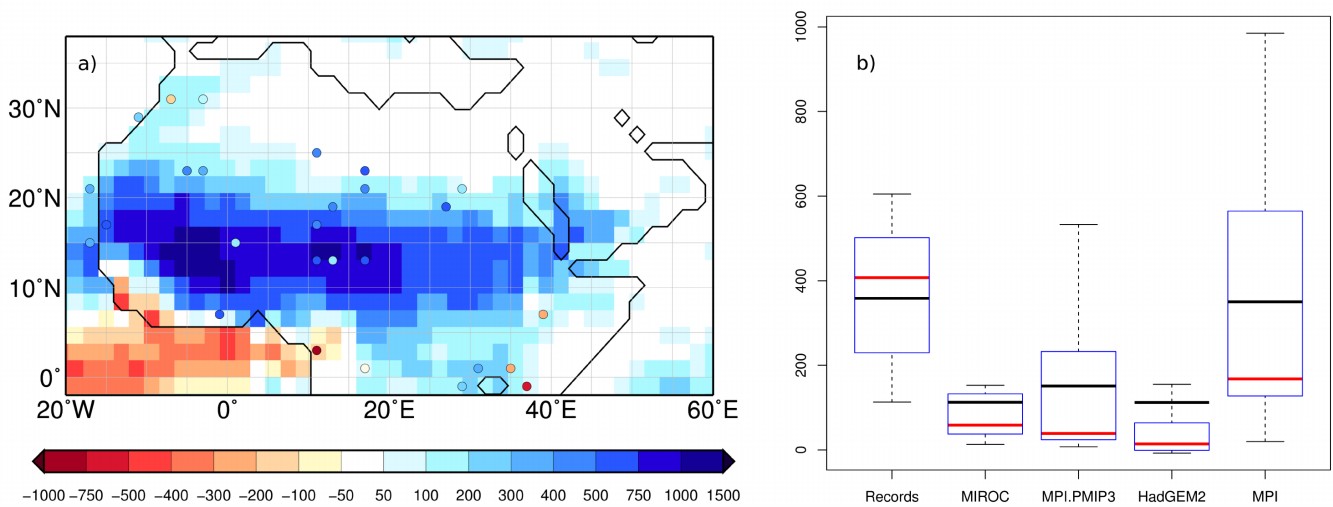

Fig. 2: **a)** Simulated (shaded) and reconstructed (dots, Bartlein et al.2011) difference in annual mean precipitation [mm/year] between 6k and 0k and **b)** boxplot of the annual mean precipitation difference [mm/year] between 6k and 0k based on all available records north of 10°N (Bartlein et al., 2011; cf. Braconnot et al., 2012) and the grid cells in the different models, in which the sites of these records are located. Considered are the PMIP3-models including dynamic vegetation (MIROC, MPI.PMIP, HadGEM2) and the model used in this study (MPI). Shown is the mean (black line) and the median (red line).

60





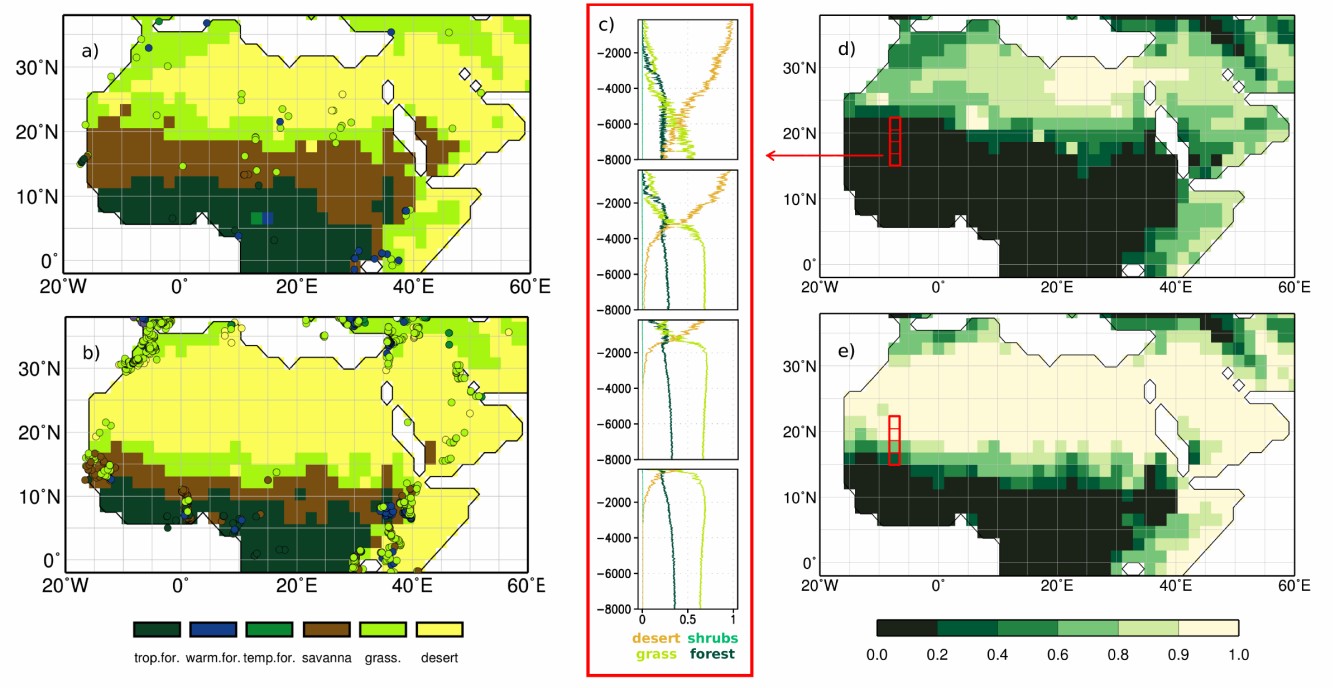

Fig. 3: **a)** Mid-Holocene (6k) and **b)** pre-industrial (0k) biome distributions based on the simulated plant functional types, following Dallmeyer et al. (2019). The reconstructed biomes (BIOME6000 project, Harrison, 2017) are displayed as dots. **c)** Simulated change in main PFT cover fraction (i.e. desert, grass, shrubs, and forest fraction) from 8k (-8000) until pre-industrial for 4 grid-cells in the western Sahel (red boxes in d and e), **d)** simulated minimum desert fraction during the Holocene (upper panel) and **e)** pre-industrial desert fraction (0k).

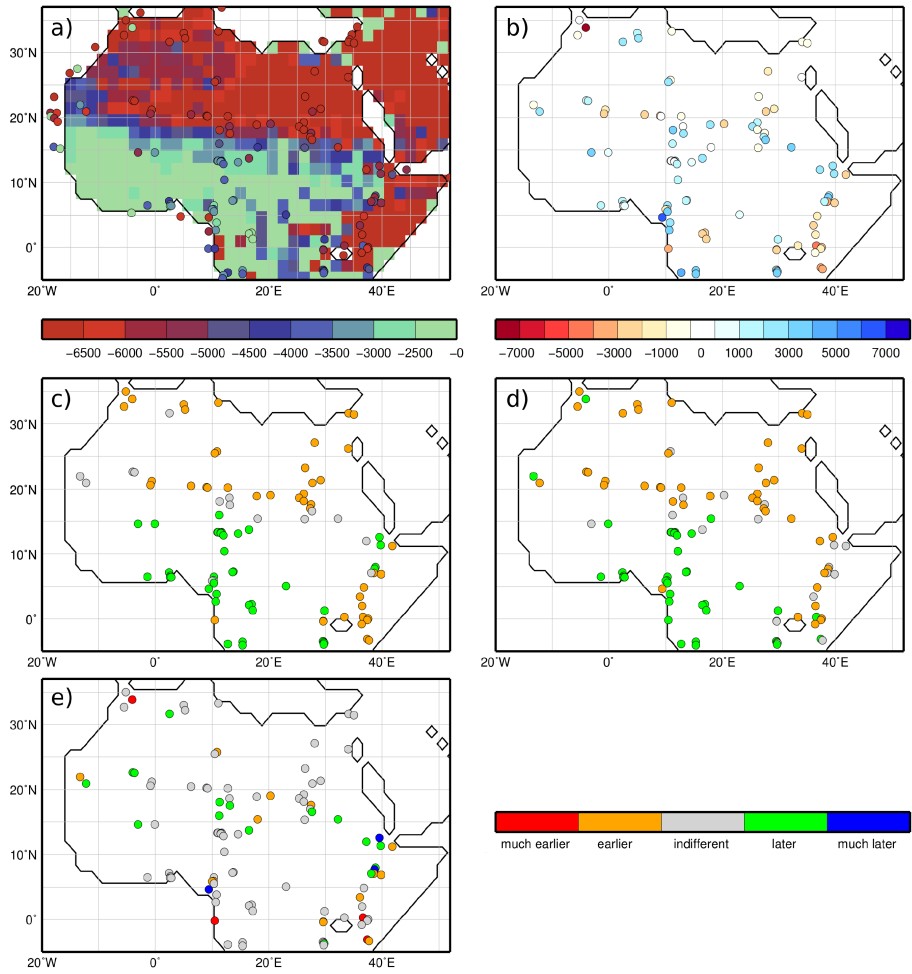

Fig. 4: **a)** Simulated end of the AHP at each individual grid-cell (shaded) and reconstructed end by Shanahan et al. (dots). **b)**: Absolute difference [years] between the AHP end in the model and in the reconstructions. **c)-e)** Comparison of the relative timing of the AHP end in the model and the reconstructions. For this, the simulated end times have been assigned to 500 yr-intervals being comparable to the temporal resolution of the records. **c)** For each grid-cell in the model in which a record site is located, it was calculated if the simulated AHP end was earlier than (orange), later than (green) or similar (gray) as to that of all other grid-cells in which sites are located. **d)** same as c) but for the records **e)** difference of the relative timing (model minus records). Please notice that for the comparison only terrestrial sites have been used.





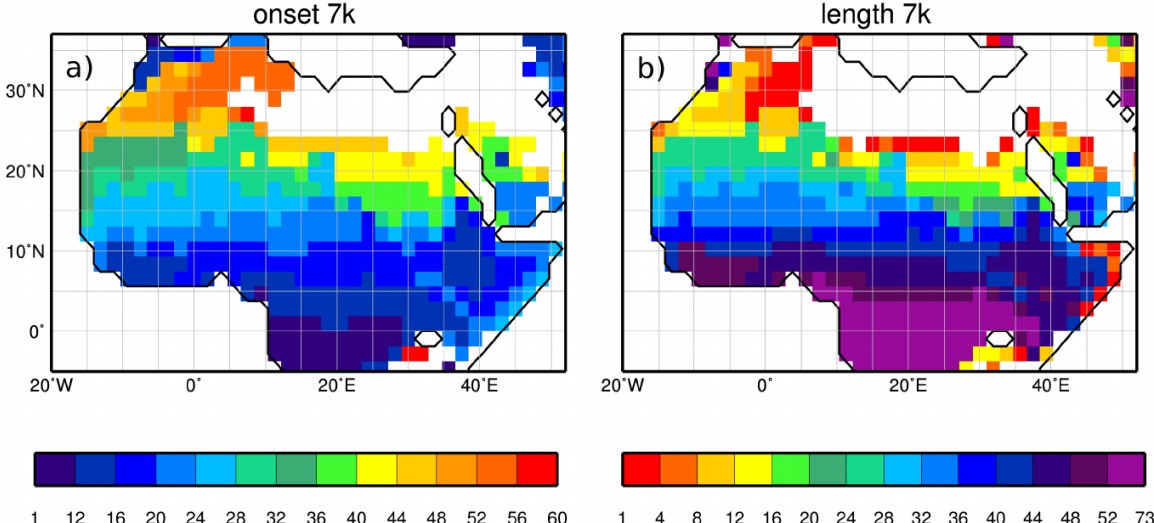

Fig. 5: Simulated **a)** monsoon onset pentad (i.e. 5-day interval) and **b)** monsoon length (in pentads) for the 7k time-slice, on land only.



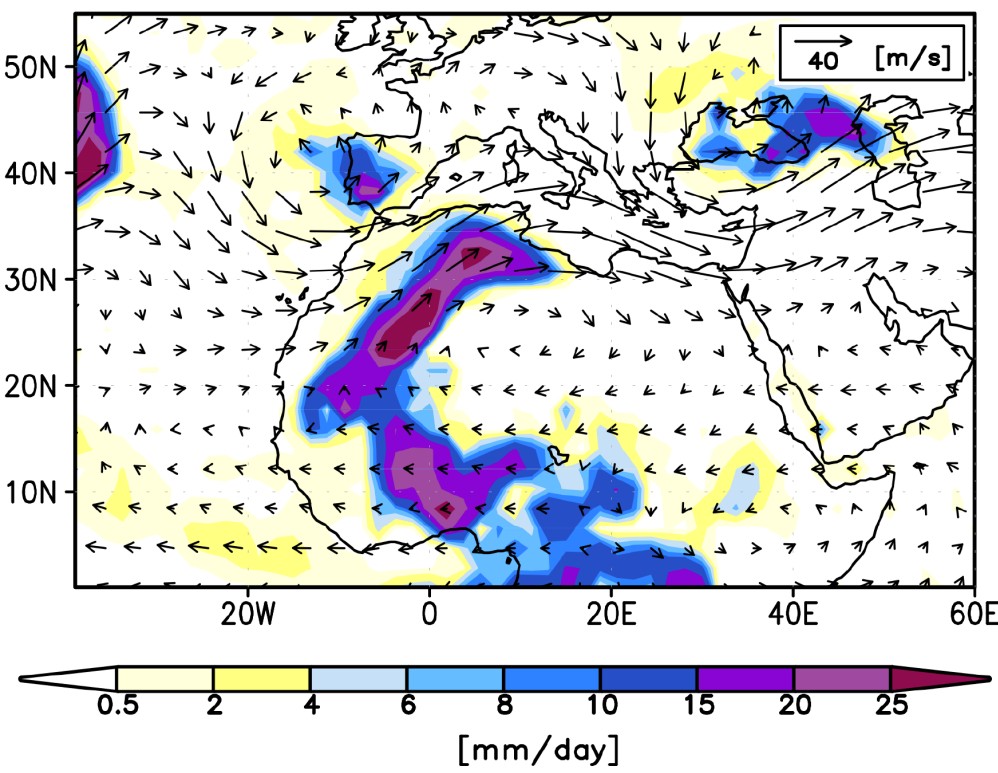

Fig. 6: Example of a strong rain event of one day in October in the Sahara at 7k, shown are the upper tropospheric wind (300hPa, vectors, in m/s) and the daily mean precipitation [mm/day].





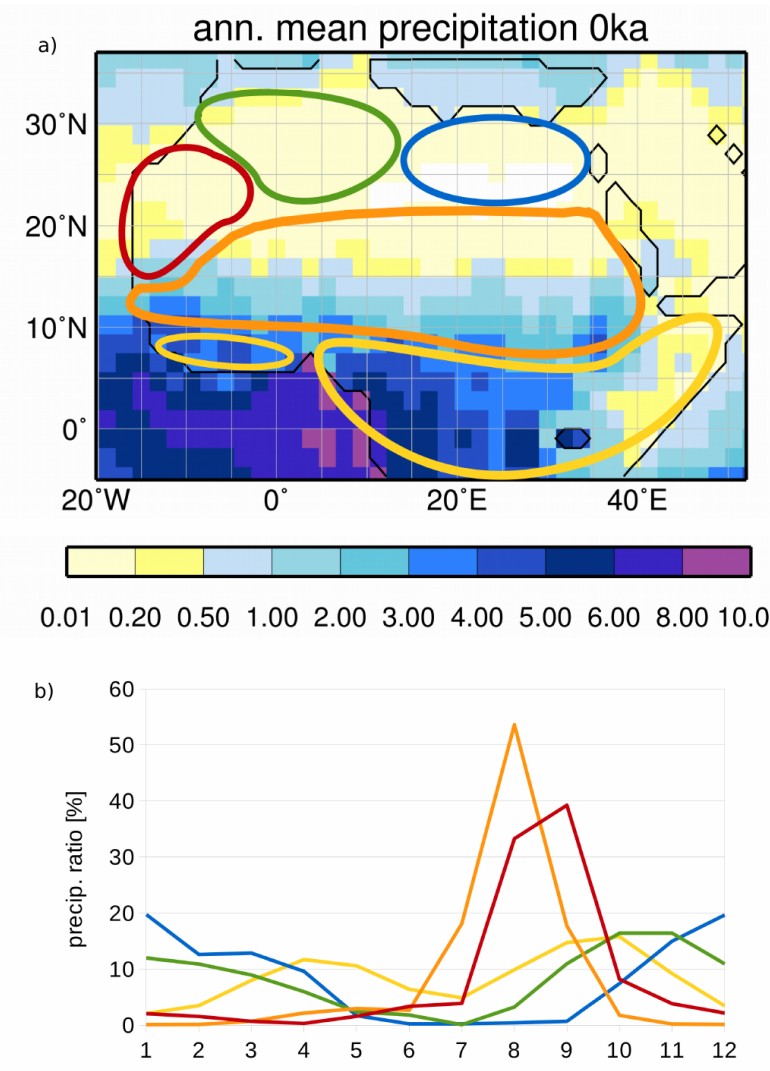

Fig. 7: **a)** Simulated pre-industrial annual mean precipitation [mm/day] and major rainfall 'regimes', **b)**: monthly pre-industrial precipitation ratio [%] in the different regimes, i.e. the ratio of monthly mean precipitation to annual mean precipitation.

70



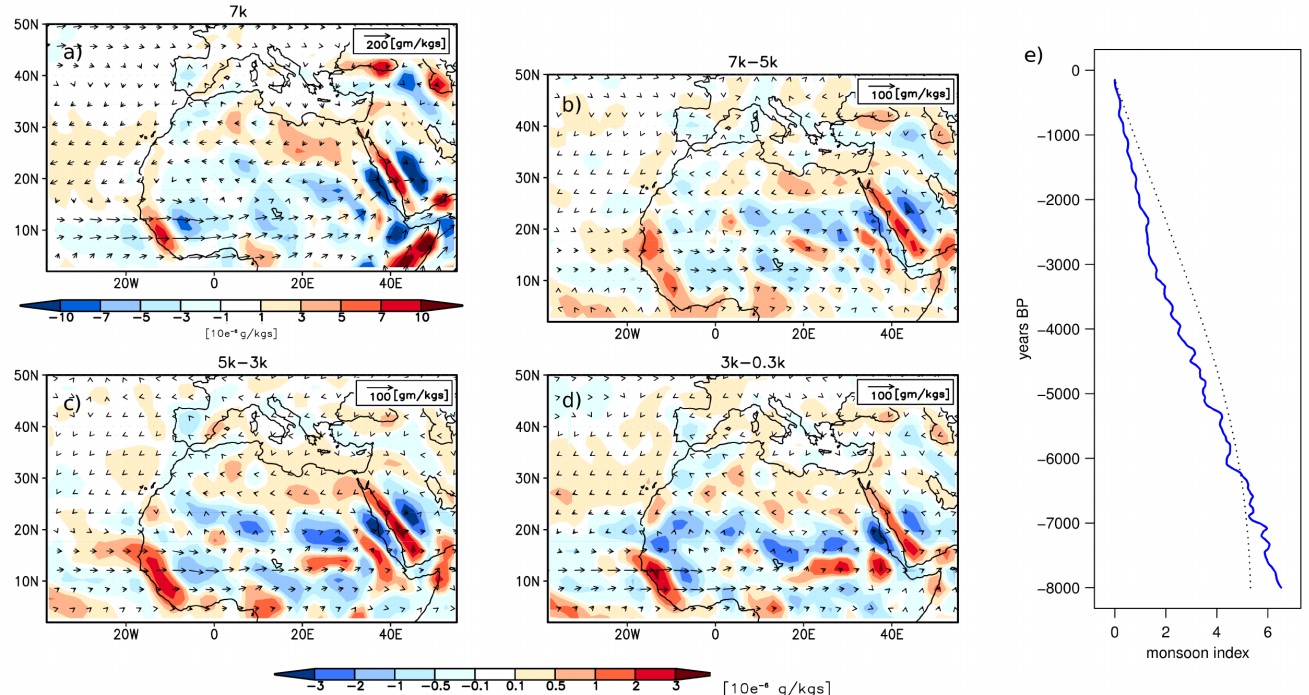

Fig. 8: **a)-d)** Moisture flux (vector, in g/kg m/s) on the 850hPa level and its divergence (shaded, blue = convergence, red = divergence, in $10e^{-5}$ g/kg m) for the months July-Sept (adjusted to fixed-angular calendar definition), for **a)** 7k and the differences **b)** 7k-5k, **c)** 5k-3k and **d)** 3k-0.3k. **e)** Smoothed change in monsoon strength, based on a simple index (q·u, averaged over the region 10W-0E,8N-18N, blue), and the Rossignol-Strick-Index (Rossignol-Strick, 1985) (black), both as relative change compared to 0k (t-0k/0k).





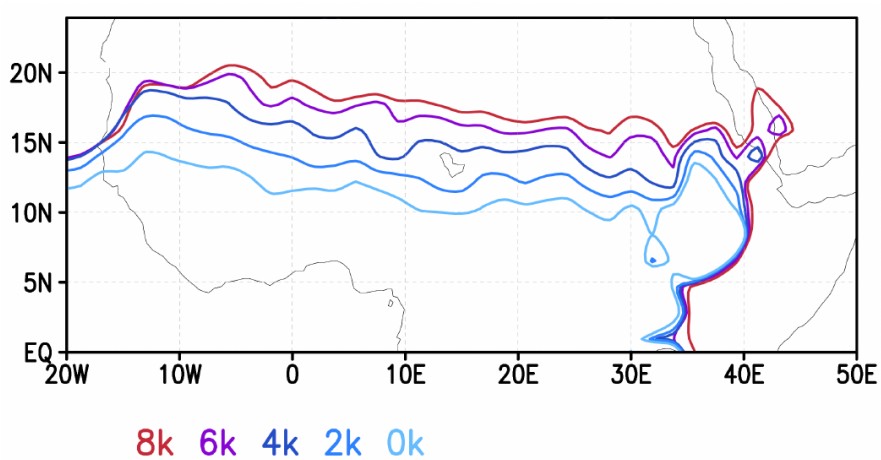

Fig. 9: Change of the northernmost extent of the monsoon rainband, based on the simulated 2mm/day annual mean precipitation isohyet.





Fig. 10: Number of rain events in September and October that exceed the 4mm/day precipitation criteria, for **a)** the 7k time-slice, and for the differences in the number of rain events (>4mm/day) between time-slices: **c)** 7k-5k, **b)** 5k-3k and **d)** 3k-0.3k.

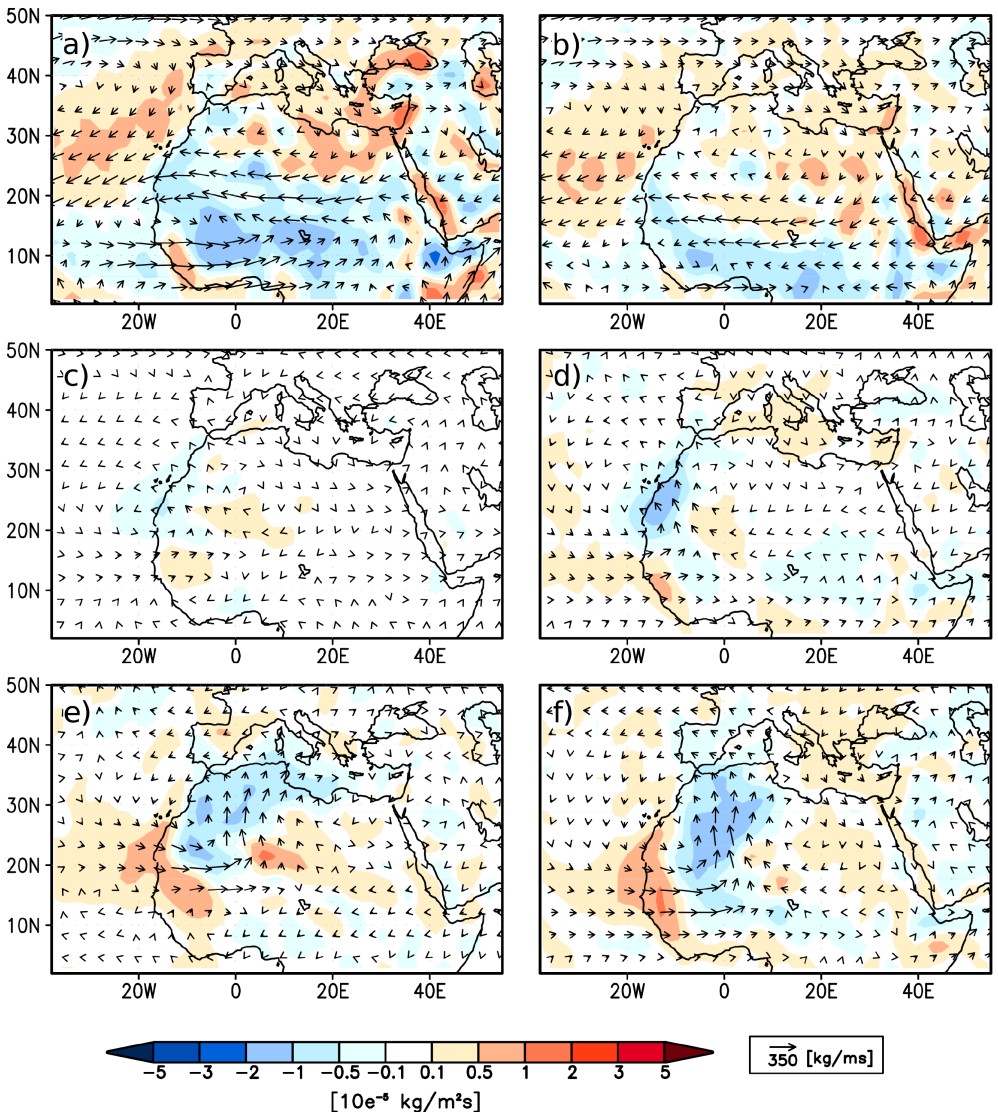

Fig. 11: Monthly mean vertically integrated moisture flux (vectors, in kg/ms) and its convergence (shaded, convergence = blue, divergence = red, in $10e^{-5}kg/m^2s$) for **a)** 7k September and **b)** 7k October (both months have been adjusted to the fixed-angular calendar); **c)-d)** differences in vertically integrated moisture flux and its convergence between the composite of all rain events occurring in the coastal area (15-11W, 20-25°N) and the monthly mean flow for **c)** 7k September, **d)** 7k October; and **e)-f)** the differences between the composite of all rain events in the Western Sahara (20-35°N, 10°W-10°E) and the monthly mean flow for **e)** 7k September and **f)** 7k October.





Fig. 12: Change in 300hPa geopotential height (m, shaded) and 300hPa wind (m/s, vector) from **a)** 7k to 5k September, **b)** 7k to 5k October, **c)** 5k to 3k September and **d)** 5k to 3k October, respectively. The months have been adjusted to the fixed-angular calendar.



Fig. 13: Change in 850hPa geopotential height (m, shaded) and 850hPa wind (m/s, vector) from **a)** 7k to 5k September, **b)** 7k to 5k October, **c)** 5k to 3k September and **d)** 5k to 3k October, respectively. The months have been adjusted to the fixed-angular calendar.

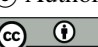

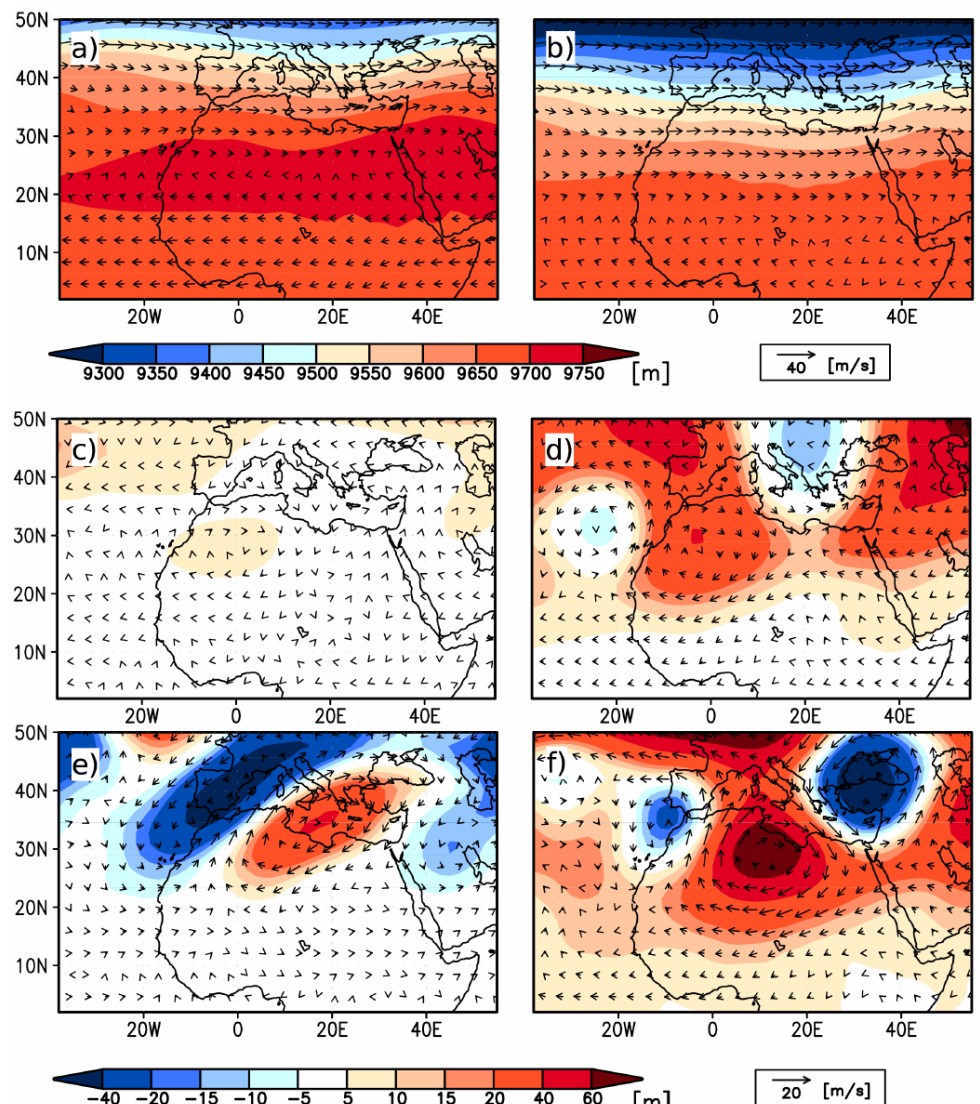

Fig. A1: Monthly mean geopotential height at 300hPa (m, shaded) and wind at 300hPa (m/s, vector) for **a)** 7k September and **b)** 7k October. **c)-d)** Differences in 300hPa geopotential height (m, shaded) and 300hPa wind (m/s, vector) between the composite of all rain events occurring in the coastal area (15-11W, 20-25°N) and the monthly mean flow for **c)** 7k September, **d)** 7k October; and **e)-f)** differences between the composite of all rain events in the Western Sahara (20-35°N, 10°W-10°E) and the monthly mean flow for **e)** 7k September and **f)** 7k October. The months have been adjusted to the fixed-angular calendar.




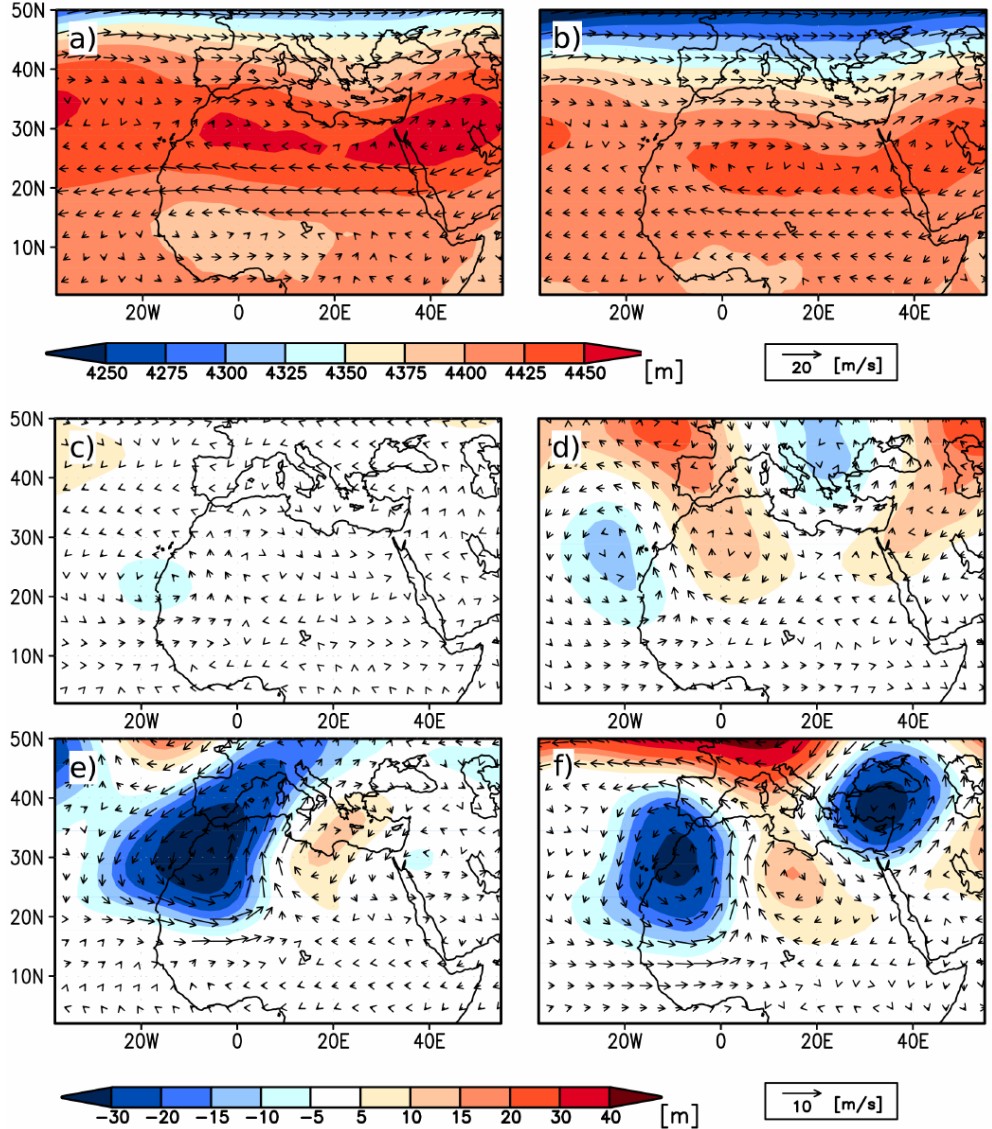

Fig. A2: Monthly mean geopotential height at 600hPa (m, shaded) and wind at 600hPa (m/s, vector) for **a)** 7k September and **b)** 7k October. **c)-d)** Differences in 600hPa geopotential height (m, shaded) and 600hPa wind (m/s, vector) between the composite of all rain events occurring in the coastal area (15-11W, 20-25°N) and the monthly mean flow for **c)** 7k September, **d)** 7k October; and **e)-f)** differences between the composite of all rain events in the Western Sahara (20-35°N, 10°W-10°E) and the monthly mean flow for **e)** 7k September and **f)** 7k October. The months have been adjusted to the fixed-angular calendar..

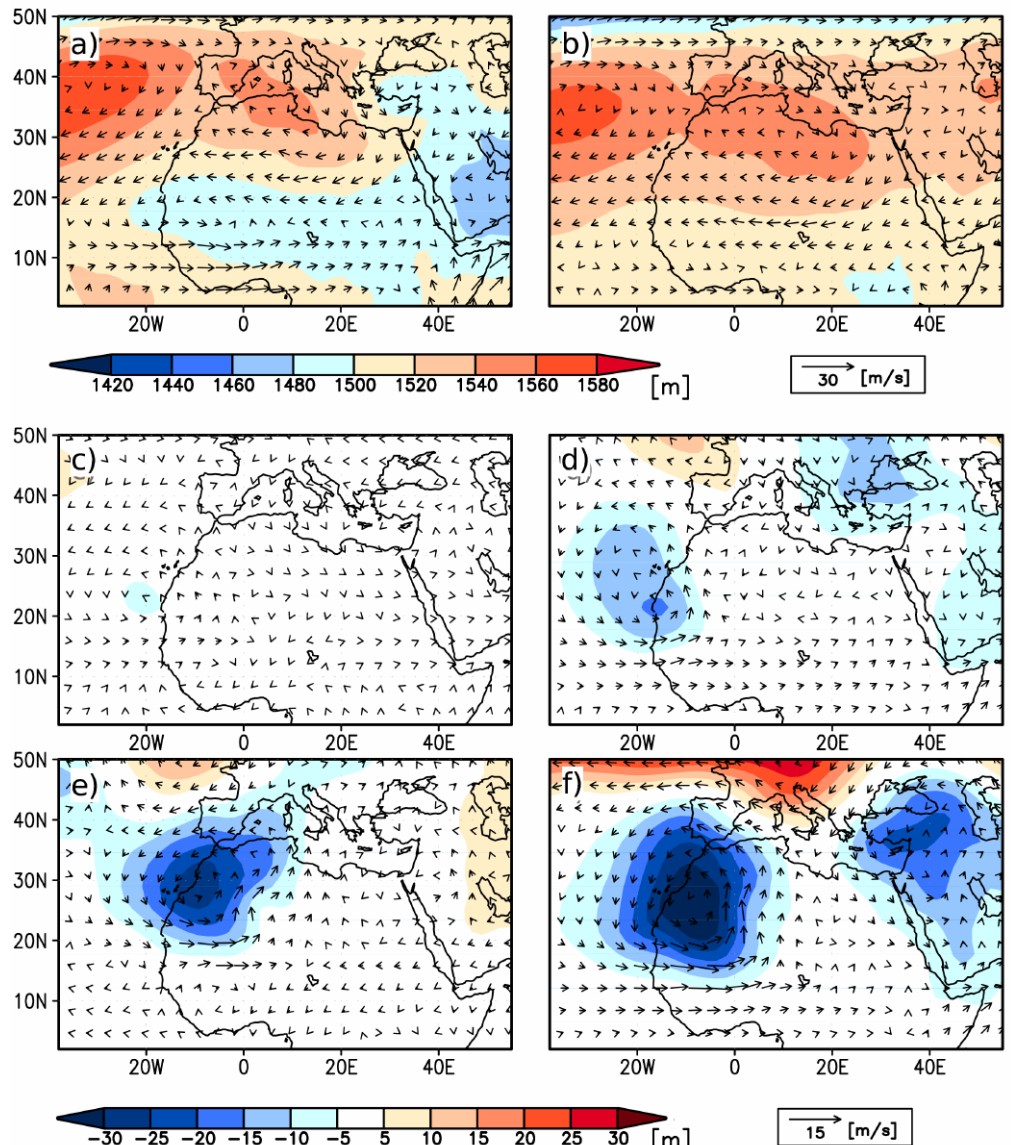

Fig. A3: Monthly mean geopotential height at 8500hPa (m, shaded) and wind at 850hPa (m/s, vector) for **a)** 7k September and **b)** 7k October. **c)-d)** Differences in 850hPa geopotential height (m, shaded) and 850hPa wind (m/s, vector) between  the composite of all rain events occurring in the coastal area (15-11W, 20-25°N) and the monthly mean flow for **c)** 7k September, **d)** 7k October; and **e)-f)** differences between the composite of all rain events in the Western Sahara (20-35°N, 10°W-10°E) and the monthly mean flow for **e)** 7k September and **f)** 7k October. The months have been adjusted to the fixed-angular calendar.