# Peer review of "The end of the African humid period as seen by a transient comprehensive Earth system model simulation of the last 8000 years"

_Climate of the Past, 2019_

## Referee Comment (RC1) · Anonymous Referee #1 · 14 Aug 2019

This paper discusses the general atmospheric circulation patterns that led to a time-transgressive termination of Green Sahara conditions in their 7850-year transient simulation with MPI-ESM1.2. The paper is well-written, carefully organized, and very detailed. The scientific questions addressed in this paper are well-motivated and the results are logically laid out and backed by supporting evidence. The findings of this paper present important conclusions and significant advances to our understanding of climate dynamics that contributed to the end of the African Humid Period.

The main result from this study is an assessment of the atmospheric circulation changes that brought about regionally diverse terminations of the African Humid Period between 8ka and the Pre-industrial era. The authors highlight that changes in West African monsoon dynamics and occurrence of extratropical troughs are the dominant mechanisms by which local regions experienced an end to humid conditions during the Holocene. The description of their transient Earth system model simulations is detailed and clear, with the exception of their description of how vegetation is treated by the model, which is covered in greater detail below.

This paper significantly contributes to the field of Holocene African climate dynamics, and I certainly recommend it for publication. However, I have listed some clarifications and modifications below that will help strengthen the quality of this publication.

Page 4, Line 110 and 3.2 Vegetation distribution: One challenge of using a transient climate simulation with dynamic vegetation during the African Humid Period is ensuring that the vegetation-climate interactions are being simulated accurately and that the vegetation does not die off or grow too rapidly. It appears that these simulations have successfully simulated African vegetation throughout the last 8ka; however, there needs to be more detail here on how vegetation is treated in the model. What was the initial condition used for vegetation in North Africa at the start of the climate simulations (at 8ka)? What are the moisture thresholds for changing between vegetation types and ultimately drying out the land surface for an end to the AHP? Vegetation is incredibly important for simulating Holocene North African hydroclimate, so it is important to add these descriptions either to the Methods or in 3.2 Vegetation distribution. These details may be present in some of your cited literature (e.g., Dallmeyer et al., 2019 or Bader et al., 2019), but I believe it necessary to include the important details here in this paper as well.

Page 4, Lines 125-128: When discussing the orbital-induced insolation changes taking place in the transient simulation, it would help for understanding of the simulation setup to explain how often these vary. Are the values changing year-to-year (so year 7001 has very slightly different values than 7002) or are they fixed values for a certain time span (i.e. step-wise changes; fixed orbital values for years 7001-7050 then different

values for 7051-7100)?

Page 4-5, 2.1 The transient simulation: In addition to added detail regarding model treatment of vegetation, it is important to understand how the model was spun-up for a simulation start at 8ka. Please add a brief description of the process undertaken to spin-up the model in Section 2.1. Again, this information may be present in Bader et al. (2019), but I believe that mention of this process is important for clarity on model setup.

Page 5, Line 143-144: I understand that the periods 7k, 5k, 3k, and 0.3k are selected as periods with low volcanic activity; however, they also seem to provide and be used as a four-part snapshot of decreasing orbital precession and North African humidity. It may be useful to add further description on the use and motivation for using these time-slices to analyze changes in atmospheric circulation throughout the transient simulations.

Page 6, Lines 166-167: The following sentence was somewhat ambiguous to me: "In a few cases, records that were deemed too short to properly identify the decline in precipitation (i.e., because of deflation), were excluded." If I understand this correctly, this sentence is describing that there are too few data points available in these records to identify a robust decline, so therefore these records were not used. If this is not the intended purpose of this sentence, please update to improve clarity.

Page 9, Lines 260-261: The phrase "In the region north of 7°N" is somewhat ambiguous for this calculation. I recommend you make this region more well-defined so it is clear where this calculation is taking place.

Page 10, Lines 316-319: The bipolar convergence pattern is somewhat difficult to discern in Figure 8, especially because the listed reference (Fig. 8a) is not a difference plot. I would recommend adding description to this section to more clearly define where this bipolar change is occurring – either by including a description of a well-defined region or by placing a dotted box in Figure 8c, which appears to be the difference plot

being referenced.

Page 14, Lines 430-434: Instead of describing that "The AHP . . . is still present in pre-industrial times", could the description instead state that parts of this equatorial region are not impacted by the orbital precession-fueled swings in monsoonal rainfall, like the other listed regions are, due to the fact that its rainfall comes from a variety of patterns, only one of which is monsoonal? This is just a suggestion that may make this section read more smoothly.

Figure 3: Showing how types of vegetation evolve in four different grid cells throughout the transient simulation is a very interesting way to analyze how the AHP termination varies! The grid cells chosen here are clustered in a fairly small meridional arrangement, however. It might be more informative to expand these grid cells to show this same vegetation time-evolution for grid cells further north and further south to provide more context of vegetational changes. However, if these do not provide novel vegetation change results, then this would be unnecessary.

Figure 3: d) shows the simulated minimum desert fraction during the Holocene, but it is unclear at what point of the simulation this distribution comes from. I would suggest that a description is added to define when this distribution occurred.

Figure 4: The explanation of c) – e) is somewhat confusing. I interpreted this as - c) Model: orange dots experience AHP end first (before all other colors) and green dots last (after all other colors) - d) Records: orange dots experience AHP end first (before all other colors) and green dots last (after all other colors) - e) Orange dots are where the Model dots were earlier than the Record dots, Green/Blue dots are where the Model dots were later than the Record dots, and grey dots are where the Model and Record dots were the same. If this interpretation is correct, one suggestion for improving clarity could be to add a different color bar for e) so the absolute and difference plots do not have the same color bar. This may make the colors of the dots more intuitive and help understanding.

Purely technical corrections are listed below:

Page 2, Line 50: "Neeling" should be changed to "Neelin", I believe.

Page 7, Line 212: There is a reference to BIOME6000, Harrison, 2017. However, I was unable to reconcile this in-text citation with the References at the end. Please update the citation in the References. Perhaps this is supposed to be Harrison, 2014? Or a paper not listed?

Page 10, Line 315: There is a reference made to "Fig. 8b" here, but the previously stated description does not appear to be in reference to what I see in Figure 8b. I believe it may be in reference to Figure 8e instead?

Page 13, Line 407: There is a reference to "Fig. A4", but this figure does not appear in the paper. Please update this reference.

---

## Referee Comment (RC2) · Anonymous Referee #2 · 26 Aug 2019

Peer Review File

Title: The end of the African humid period as seen by a transient comprehensive Earth system model simulation of the last 8000 years Authors: Ann Dallmeyer, Martin Claussen, Stephan J. Lorenz, and Timothy Shanahan

Summary The authors present analysis of the African Humid Period (AHP) in a transient earth system model simulation of the mid to late Holocene. They focus on characterizing and understanding the termination of the AHP (wet to dry transition) across northern Africa. They show that wet mid-Holocene conditions were primarily confined to western and central regions of the Sahara, and that the transition to present-day aridity in these regions was time transgressive, consistent with proxy interpretation. Based on analysis of modeled daily precipitation events and subtropical jet stream characteristics, they find that tropical-extratropical interactions in the form of tropical plumes enhanced mid-Holocene rainfall in western regions of the Sahara, prolonging wet conditions there. The tropical plume hypothesis helps to explain the spatial differences in AHP termination date across the Sahara.

The paper adds to our understanding of the AHP in two important ways. First, it confirms earlier hypotheses and simplified modeling results that tropical-extratropical interactions shaped the AHP using an advanced, state of the art coupled climate model simulation. Second, it presents the tropical-extratropical interaction mechanism in the context of the spatially and temporally heterogeneous termination of the AHP. The analysis is well done and straightforward and the manuscript is well-written. I recommend the manuscript be considered for publication in Climate of the Past, and only have a few minor comments for the authors.

Minor comments

Please discuss the suitability of the T63 resolution for studying tropical plumes. Can MPI-ESM1.2 accurately simulate these fairly narrow, transient events?

There are several instances when the authors reference Skinner and Poulsen (2013), but the reference should be Skinner and Poulsen (2016).

Lines 148-149: Please provide evidence (references) that the use of the bare soil fraction is an appropriate indicator for moisture availability in the Sahara. I imagine that this depends strongly on the dynamic vegetation module.

Can you provide a discussion of why the Eastern Sahara does not see a substantive increase in precipitation like the Western Sahara in MPI? Why does the monsoon enhancement remain constrained to the west? This is the opposite of what we see in CMIP5 projections for the 21st century in response to elevated GHG forcing, so it may

have relevance for understanding future climate.

Line 407: The authors reference a Figure A4, but it was not included in the draft.
* * *

---

## Author Comment (AC1) · 18 Oct 2019

We thank Referee#1 for his useful, constructive comments that have helped to strengthen the manuscript.

R: This paper discusses the general atmospheric circulation patterns that led to a time-transgressive termination of Green Sahara conditions in their 7850-year transient simulation with MPI-ESM1.2. The paper is well-written, carefully organized, and very detailed. The scientific questions addressed in this paper are well-motivated and the results are logically laid out and backed by supporting evidence. The findings of this paper present important conclusions and significant advances to our understanding

of climate dynamics that contributed to the end of the African Humid Period. The main result from this study is an assessment of the atmospheric circulation changes that brought about regionally diverse terminations of the African Humid Period between 8ka and the Pre-industrial era. The authors highlight that changes in West African monsoon dynamics and occurrence of extratropical troughs are the dominant mechanisms by which local regions experienced an end to humid conditions during the Holocene. The description of their transient Earth system model simulations is detailed and clear, with the exception of their description of how vegetation is treated by the model, which is covered in greater detail below. This paper significantly contributes to the field of Holocene African climate dynamics, and I certainly recommend it for publication. However, I have listed some clarifications and modifications below that will help strengthen the quality of this publication.

R: Page 4, Line 110 and 3.2 Vegetation distribution: One challenge of using a transient climate simulation with dynamic vegetation during the African Humid Period is ensuring that the vegetation-climate interactions are being simulated accurately and that the vegetation does not die off or grow too rapidly. It appears that these simulations have successfully simulated African vegetation throughout the last 8ka; however, there needs to be more detail here on how vegetation is treated in the model. What was the initial condition used for vegetation in North Africa at the start of the climate simulations (at 8ka)? What are the moisture thresholds for changing between vegetation types and ultimately drying out the land surface for an end to the AHP? Vegetation is incredibly important for simulating Holocene North African hydroclimate, so it is important to add these descriptions either to the Methods or in 3.2 Vegetation distribution. These details may be present in some of your cited literature (e.g., Dallmeyer et al., 2019 or Bader et al., 2019), but I believe it necessary to include the important details here in this paper as well.

A: We agree with the Referee that including an introduction of the dynamic vegetation model will help to understand the vegetation change in Africa. We split the method section on the transient simulation in "2.1 The model MPI-ESM1.2" and "2.2 The transient simulation" and added the following information to these sections: regarding the vegetation description in the model: "Natural vegetation is represented by eight different plant functional types (tropical or temperate evergreen or deciduous trees, respectively, raingreen and cold resistant shrubs, C3 and C4 grass) which can in principle coexist in each grid-cell as the model uses a tiling approach. The occurrence of each PFT is constrained by temperature thresholds representing their respective bioclimatic tolerance. The fractional cover of each PFT is by and large determined by the relative differences in annual net primary productivity (NPP) between the PFTs which - among other factors – depends on the moisture ability and requirement of the plants. The establishment of PFTs is furthermore reduced by disturbances and weighted by the inverse of the PFT-specific lifetime. Woody PFTs are generally favoured over grass, but in regions with frequent disturbances or bioclimatic conditions near the thresholds, shrubs or even grass may win the competition as they can recover more quickly than trees. For each grid cell, a bare soil fraction (BSF) is considered in addition to the vegetated area, which represents the seasonal and permanently unvegetated ground. Their fraction is calculated via the relation of maximum carbon storage in the pool for living tissues to the carbon actually stored in this pool by the NPP, representing the need of plants of a certain amount of carbon to build up their leafs, etc. so that they can function properly. If the filling of the pool is not sufficient for all PFTs, plants cannot grow and the grid-cell is mainly non-vegetated. Thus, simulated changes in vegetation cover can be attributed to bioclimatic shifts (i.e. temperature changes), changes in plant productivity (related to precipitation) or changes in the frequency of disturbances. More details and information about the dynamic vegetation module is given in Brovkin et al. (2009) and Reick et al.(2013). "

Regarding the initial conditions of the vegetation: We are aware of the fact that it is still being discussed whether multiple climate and vegetation states are possible for North Africa. Due to previous simulations in similar model setups as used here, we rather would expect multiple states for the pre-industrial time-slice, but not for the early

mid-Holocene. Therefore, we have not tested the dependence of the timing of the AHP end on the initial condition in the vegetation. The model started from pre-industrial vegetation and were ran into quasi-equilibrium with fixed boundary conditions for the 8k time-slice (see comment below) and for this quasi-equilibrium state, the model simulates a 'green' Sahara. If there were an even greener solution for the initial vegetation (i.e. starting the transient simulation from a fully vegetated state), we would expect an even more rapid end of the AHP and a change in the timing, but not in the pattern and the relative timing of the AHP end. We added to the description of the transient simulation: "The dependence of the AHP end on the initial vegetation conditions was not tested. However, based on previous simulations in similar model versions, we do not expect multiple vegetation conditions for the Sahara of the early Mid-Holocene, so that the results of this study are assumed to be independent of the initial condition. Nevertheless, it should be noted that initial conditions and model setup may have an impact on transient simulations and their interpretation (c.f. Braconnot et al., 2019)."

Reference: Braconnot, P., Zhu, D., Marti, O., and Servonnat, J.: Strengths and challenges for transient Mid- to Late Holocene simulations with dynamical vegetation, Clim. Past, 15, 997–1024, https://doi.org/10.5194/cp-15-997-2019, 2019.

Regarding the moisture limits: JSBACH does not restrict the area that can be covered by PFTs by moisture limits. Precipitation and the available moisture affect the vegetation by having an impact on the NPP which in turn affects the storage of carbon in the pool of living tissues by which the bare soil fraction is calculated. We added the following lines to the vegetation model description: "The fractional cover of each PFT is by and large determined by the relative differences in annual net primary productivity (NPP) between the PFTs which - among other factors – depends on the moisture ability and requirement of the plants. "

R: Page 4, Lines 125-128: When discussing the orbital-induced insolation changes taking place in the transient simulation, it would help for understanding of the simulation setup to explain how often these vary. Are the values changing year-to-year (so

year 7001 has very slightly different values than 7002) or are they fixed values for a certain time span (i.e. step-wise changes; fixed orbital values for years 7001-7050 then different values for 7051-7100)?

A: To clarify the variation of the boundary conditions, we inserted the following section after explaining the forcing mechanisms: "The slowly evolving orbital parameters (a) and smoothed greenhouse gases (b) are updated for every decade. The other forcing mechanisms, sulfate aerosols (c), SSI (d), and land-use data are read annually and calculated daily by linear interpolation. A detailed description on the transient simulation and the forcing mechanisms is given in ..."

R: Page 4-5, 2.1 The transient simulation: In addition to added detail regarding model treatment of vegetation, it is important to understand how the model was spun-up for a simulation start at 8ka. Please add a brief description of the process undertaken to spin-up the model in Section 2.1. Again, this information may be present in Bader et al. (2019), but I believe that mention of this process is important for clarity on model setup.

A: We agree and we added the following information to the method section: "We conducted a "spin-down" simulation to capture the model's response to constant boundary conditions of the mid-Holocene climate. For this "spin-down" simulation, the model has been started from pre-industrial climate and vegetation conditions, and the external forcing mechanisms were kept constant to the values of the year 6000 BCE. The model ran more than 1000 years to reach quasi-equilibrium between the boundary conditions, climate and the carbon cycle. The transient simulation started from this equilibrium state and was run until pre-industrial time (i.e. 1850 CE)."

R:Page 5, Line 143-144: I understand that the periods 7k, 5k, 3k, and 0.3k are selected as periods with low volcanic activity; however, they also seem to provide and be used as a four-part snapshot of decreasing orbital precession and North African humidity. It may be useful to add further description on the use and motivation for

using these time-slices to analyze changes in atmospheric circulation throughout the transient simulations.

A: For the analysis of the extratropical trough activity, daily output is needed. Since the output of such a transient simulation can become very large, only monthly mean values of the main simulation were stored. For the daily values the model was therefore restarted for very short transient simulations. The time slices were chosen relatively arbitrarily, the main criterion was a low variability in volcanic activity. We modified the sentences to: "For this purpose, a few 30-year long time-slice experiments were re-run that represent snapshots of decreasing orbital precession and North African humidity. For these time-slices, periods with low volcanic activity were selected. Here we chose periods around 7k, 5k, 3k and 0.3k (Table.1)."

R: Page 6, Lines 166-167: The following sentence was somewhat ambiguous to me: "In a few cases, records that were deemed too short to properly identify the decline in precipitation (i.e., because of deflation), were excluded." If I understand this correctly, this sentence is describing that there are too few data points available in these records to identify a robust decline, so therefore these records were not used. If this is not the intended purpose of this sentence, please update to improve clarity.

A: It is not only the low temporal resolution of records, but also the fact that some records end or have gaps, so that the end of the African humid period cannot be determined accurately enough. We wrote: "A few records that are affected by deflation or have a too low temporal resolution and therefore do not represent the decrease in precipitation accurately enough, were excluded because the end of the AHP could not properly be identified in these records"

R: Page 9, Lines 260-261: The phrase "In the region north of 7°N" is somewhat ambiguous for this calculation. I recommend you make this region more well-defined so it is clear where this calculation is taking place.

A: We re-do this analysis with a more precise area and write: "In the monsoon affected

region outside the area with perennial rainfall (i.e. the continental area between 7.46-37.21N, 19.69-53.44E), the end of the AHP is negatively correlated with the onset of the rainy season at 7k (r=-0.56) and strongly positively correlated with the rainy season length (r=0.61). "

R: Page 10, Lines 316-319: The bipolar convergence pattern is somewhat difficult to discern in Figure 8, especially because the listed reference (Fig. 8a) is not a difference plot. I would recommend adding description to this section to more clearly define where this bipolar change is occurring – either by including a description of a well-defined region or by placing a dotted box in Figure 8c, which appears to be the difference plot being referenced.

A:Thank you for remarking this. The references of the figures were not correct (it is the entire figure 8, not figure 8a), we changed this. For clarification we added a cyan line in the difference plots, seperating the divergence and convergence and also added a description of this cyan line to the figure caption.

R: Page 14, Lines 430-434: Instead of describing that "The AHP . . . is still present in preindustrial times", could the description instead state that parts of this equatorial region are not impacted by the orbital precession-fueled swings in monsoonal rainfall, like the other listed regions are, due to the fact that its rainfall comes from a variety of patterns, only one of which is monsoonal? This is just a suggestion that may make this section read more smoothly.

A: We agree with the Referee. The regional climate dynamics are complex and the monsoon circulation may not be the only explanation. A detailed analysis of the climate-vegetation relationships would go beyond this publication. We added to the revised version: "This contrasting trend may partly be related to the fact, that rainfall in western equatorial Africa is the result of complex regional and remote interactions between the tropical oceans, the orography and the atmospheric circulation. In present-day observation, the region shows the highest rainfall rates in North Africa and

maximum moisture recycling ratios. The seasonal precipitation cycle is related to the insolation changes, but the monsoonal cycle explains only parts of the regional climate variability (cf. Dezfuli, 2017 and references therein)."

Reference: Dezfuli, A., 2017: Climate of Western and Central Equatorial Africa. Oxford Research Encyclopedia of Climate Science. https://doi.org/10.1093/acrefore/9780190228620.013.511.

R: Figure 3: Showing how types of vegetation evolve in four different grid cells throughout the transient simulation is a very interesting way to analyze how the AHP termination varies! The grid cells chosen here are clustered in a fairly small meridional arrangement, however. It might be more informative to expand these grid cells to show this same vegetation time-evolution for grid cells further north and further south to provide more context of vegetational changes. However, if these do not provide novel vegetation change results, then this would be unnecessary.

A: The grid-cells chosen here are just an example on how the transient vegetation change takes place in the model. We added a map showing the transient change in main PFTs for all grid-cells in North Africa to the supplement.

R:Figure 3: d) shows the simulated minimum desert fraction during the Holocene, but it is unclear at what point of the simulation this distribution comes from. I would suggest that a description is added to define when this distribution occurred.

A:This is an interesting aspect indeed, but we think that it would not be relevant for the understanding of the study. In the routine for calculating the AHP end it is implemented as a condition that the minimum of the desert fraction must have already been passed at the time of the AHP end. In most regions the maximum is before 6k, exception is the part of West Africa south of about 15°N where the minimum occurs later. We did not add a map showing the time of minimum desert fraction, as we provided the plot with transient vegetations changes in each grid-box. Based on this plot, the minimum in desert fraction can be accessed for each grid-cell.

R: Figure 4: The explanation of c) – e) is somewhat confusing. I interpreted this as – c) Model: orange dots experience AHP end first (before all other colors) and green dots last (after all other colors) - d) Records: orange dots experience AHP end first (before all other colors) and green dots last (after all other colors) - e) Orange dots are where the Model dots were earlier than the Record dots, Green/Blue dots are where the Model dots were later than the Record dots, and grey dots are where the Model and Record dots were the same. If this interpretation is correct, one suggestion for improving clarity could be to add a different color bar for e) so the absolute and difference plots do not have the same color bar. This may make the colors of the dots more intuitive and help understanding.

A: We agree, the explanation and the colorbar is misleading. We changed the caption and the colorbar and wrote: "e) difference of the relative timing (model minus records), e.g. 'much earlier' (red) indicates sites at which the relative time based on the model results was classified at a much earlier point in time than based on the records"

technical corrections R: Page 2, Line 50: "Neeling" should be changed to "Neelin", I believe. A: Yes, thank you. done

R: Page 7, Line 212: There is a reference to BIOME6000, Harrison, 2017. However, I was unable to reconcile this in-text citation with the References at the end. Please update the citation in the References. Perhaps this is supposed to be Harrison, 2014? Or a paper not listed? A: The reference is: Harrison, S.: BIOME 6000 DB classified plotfile version 1. University of Reading. Dataset. http://dx.doi.org/10.17864/1947.99 , 2017.

R: Page 10, Line 315: There is a reference made to "Fig. 8b" here, but the previously stated description does not appear to be in reference to what I see in Figure 8b. I believe it may be in reference to Figure 8e instead? A: Thank you, it is indeed Figure 8e.

R: Page 13, Line 407: There is a reference to "Fig. A4", but this figure does not appear

in the paper. Please update this reference. A: We changed this reference to Fig.10

---

## Author Comment (AC2) · 18 Oct 2019

We thank Referee#2 for carefully reading the manuscript and helpful comments that improve the manuscript.

Summary: The authors present analysis of the African Humid Period (AHP) in a transient earth system model simulation of the mid to late Holocene. They focus on characterizing and understanding the termination of the AHP (wet to dry transition) across northern Africa. They show that wet mid-Holocene conditions were primarily confined to western and central regions of the Sahara, and that the transition to present-day aridity in these regions was time transgressive, consistent with proxy interpretation. Based

on analysis of modeled daily precipitation events and subtropical jet stream characteristics, they find that tropical-extratropical interactions in the form of tropical plumes enhanced mid-Holocene rainfall in western regions of the Sahara, prolonging wet conditions there. The tropical plume hypothesis helps to explain the spatial differences in AHP termination date across the Sahara. The paper adds to our understanding of the AHP in two important ways. First, it confirms earlier hypotheses and simplified modeling results that tropical-extratropical interactions shaped the AHP using an advanced, state of the art coupled climate model simulation. Second, it presents the tropical-extratropical interaction mechanism in the context of the spatially and temporally heterogeneous termination of the AHP. The analysis is well done and straightforward and the manuscript is well-written. I recommend the manuscript be considered for publication in Climate of the Past, and only have a few minor comments for the authors.

Minor comments

R: Please discuss the suitability of the T63 resolution for studying tropical plumes. Can MPI-ESM1.2 accurately simulate these fairly narrow, transient events?

A: In T63 (i.e. approx.1.875° on a gaussian grid), clouds can not be resolved. Cumulus convection is parametrized. Thus, tropical plumes - in their original meaning – may not be accurately represented in the model, but the typical atmospheric circulation and the precipitation pattern associated with tropical plume events can be simulated. To demonstrate this, we have included and discussed Fig.6 in the original paper which shows a strong event in the model simulation.

R: There are several instances when the authors reference Skinner and Poulsen (2013), but the reference should be Skinner and Poulsen (2016).

A: Thank you. This was indeed a mistake.

R: Lines 148-149: Please provide evidence (references) that the use of the bare soil fraction is an appropriate indicator for moisture availability in the Sahara. I imagine that

this depends strongly on the dynamic vegetation module.

A: Following the suggestion of Referee #1, we included a short description of the vegetation dynamics in JSBACH in the revised version of the manuscript. There, we explained the calculation of the BSF, which strongly depends on the NPP of the PFTs, which in turn is a function of the moisture availability.

R: Can you provide a discussion of why the Eastern Sahara does not see a substantive increase in precipitation like the Western Sahara in MPI? Why does the monsoon enhancement remain constrained to the west? This is the opposite of what we see in CMIP5 projections for the 21st century in response to elevated GHG forcing, so it may have relevance for understanding future climate.

A: In our simulation, the precipitation enhancement during Mid-Holocene is not constrained to the Western Sahara. The precipitation and vegetation cover is also increased in the Eastern Sahara, except for the Libyan Sand Sea which presumably remained a desert in the mid-Holocene according to reconstructions. The western part experiences a much stronger increase in precipitation and vegetation which can be explained by the fact, that the West African monsoon system is mainly active in the western part of North Africa, for instance, the Saharan heat low is located in the western Sahara. The isohyets are strongly declined in North-west to South-east direction. Furthermore, the orbital forcing and the GHG forcing differ, leading to different response pattern (see e.g. Claussen et al. 2003 or D'Agostino et al. 2019).

Claussen M., Brovkin V., Ganopolski A., Kubatzki C. & Petoukhov V. (2003): Climate change in northern Africa: the past is not the future. Climatic Change, 57 (1), 99 –118.

D'Agostino, R., Bader J., Bordoni S., Ferreira D., Jungclaus J. (2019), Northern Hemisphere monsoon response to mid-Holocene orbital forcing and greenhouse gas-induced global warming, Geophys. Res. Lett. doi: 10.1029/2018gl081589.

R: Line 407: The authors reference a Figure A4, but it was not included in the draft.

A: We changed this reference to Fig.10.